# Targeting USP2 regulation of VPRBP-mediated degradation of p53 and PD-L1 for cancer therapy

Jingjie Yi [1], Omid Tavana[1], Huan Li[1], Donglai Wang[1], Richard J. Baer[1,2] & Wei Gu [1,2] ✉

Since Mdm2 (Mouse double minute 2) inhibitors show serious toxicity in clinic studies, different approaches to achieve therapeutic reactivation of p53-mediated tumor suppression in cancers need to be explored. Here, we identify the USP2 (ubiquitin specific peptidase 2)-VPRBP (viral protein R binding protein) axis as an important pathway for p53 regulation. Like Mdm2, VPRBP is a potent repressor of p53 but VPRBP stability is controlled by USP2. Interestingly, the USP2-VPRBP axis also regulates PD-L1 (programmed death-ligand 1) expression. Strikingly, the combination of a small-molecule USP2 inhibitor and anti-PD1 monoclonal antibody leads to complete regression of the tumors expressing wild-type p53. In contrast to *Mdm2*, knockout of *Usp2* in mice has no obvious effect in normal tissues. Moreover, no obvious toxicity is observed upon the USP2 inhibitor treatment in vivo as Mdm2-mediated regulation of p53 remains intact. Our study reveals a promising strategy for p53-based therapy by circumventing the toxicity issue.

*TP53* is well established as the mostly commonly mutated driver gene of human cancers[1,2]. Interestingly, however, the tumor suppression activity of the p53 pathway is also impaired through a variety of other mechanisms in many human tumors that retain a wild-type *TP53* gene[3–5]. Thus, the restoration of p53 function remains an important objective for treating human cancers with wild-type *TP53*. A popular approach has been to inhibit Mdm2, the main ubiquitin E3 ligase that normally binds p53 and downregulates its function[6]. Indeed, early work established that small-molecule antagonists of the Mdm2-p53 interaction are effective in reactivating p53 tumor suppressor function in pre-clinical models[7–9], and a variety of highly potent Mdm2-p53 antagonists (also called Mdm2 inhibitors) have since been developed and validated in vitro[10]. However, these Mdm2 inhibitors have not proven very effective in clinical trials, primarily due to their dose-limiting toxicities to normal tissues[10–17]. The adverse effects including myelosuppression, gastrointestinal symptoms, weight loss, fatigue, and cardiovascular toxicities were reported in numerous studies[11,14–22]. Treatment-related death with an Mdm2 inhibitor has also been reported in the patient with AML[14].

The severe toxicity of Mdm2 inhibitors in clinical settings reflects the reciprocal relationship between Mdm2 and p53[23–26]. On one hand, upon binding to p53, the Mdm2 protein can repress its transcriptional activity and also target it for ubiquitin-mediated degradation. On the other hand, acting as a transcription factor, p53 can bind to the *mdm2* promoter and stimulate its expression. This duality creates a negative feedback loop that tightly regulates p53 levels, allowing for rapid termination of the p53 response when p53 activity is no longer needed or becomes harmful to normal cell homeostasis[27]. Thus, Mdm2 inhibition is a two-edged sword for cancer patients. Although Mdm2 inhibitors can reactivate p53-mediated tumor suppression by disrupting the p53-Mdm2 feedback loop in tumor cells, it also can induce severe toxicities due to unleashed p53 activity in normal tissues[28]. Interestingly, however, enhanced p53 function does not always cause toxicities. For example, "Super-p53" mice that carrying extra copies of the *Tp53* gene exhibit enhanced p53 responses, including heightened protection

[1]Institute for Cancer Genetics, and Herbert Irving Comprehensive Cancer Center, Vagelos College of Physicians & Surgeons, Columbia University, 1130 Nicholas Ave, New York, NY 10032, USA. [2]Department of Pathology and Cell Biology, Vagelos College of Physicians & Surgeons, Columbia University, 1130 Nicholas Ave, New York, NY 10032, USA. ✉e-mail: wg8@cumc.columbia.edu

from tumor development in vivo[29]. Nevertheless, in contrast to Mdm2-knockout mice or mice treated with Mdm2 inhibitors, Super-p53 mice develop normally, with no obvious signs of toxicity[30]. Of note, Mdm2-mediated control of p53 remains intact and p53 protein levels are properly degraded by Mdm2 in the normal tissues of Super-p53 mice[29,30]. Moreover, by using a genetically engineered mouse with mutated p53 response elements in the *P2* promoter of *Mdm2*, it was reported that Mdm2-mediated regulation of p53 is significantly compromised in the *Mdm2*[P2/P2] mice under DNA damage conditions. Indeed, enhanced p53-dependent apoptosis upon DNA damage turns catastrophic for the integrity of the hematopoietic system, causing drastic myeloablation and lethality[31]. These data clearly demonstrate that retaining the normal Mdm2-mediated regulation is essential for the survival of stress-sensitive tissues upon p53 activation. Together, these studies suggest that the toxicities associated with the treatment of Mdm2 inhibitors are caused by deregulated p53 activation when Mdm2-mediated regulation of p53 is disrupted in normal tissues and that other approaches to achieve therapeutic reactivation of p53-mediated tumor suppression in cancer patients should be explored.

p53 is controlled by multiple pathways upon the various stress signals[32–34]. It is well established that the activity of p53 is dynamically regulated by acetylation and deacetylation[35–43]. We recently identified acidic domain-containing cofactors acting as a "reader" for unacetylated p53[44–46]. Indeed, VPRBP, an acidic domain-containing corepressor, directly interacts with the C-terminal domain and effectively suppresses p53-mediated transcription[45,47]. Of note, VPRBP is also involved in degradation of p53 by the ubiquitylation pathway, which has been validated in vivo by using VPRBP knockout mice[48]. Thus, like Mdm2, VPRBP regulates p53 functions through both transcriptional repression and ubiquitylation-mediated degradation, but it does so in an Mdm2-independent manner. Moreover, since VPRBP is overexpressed in several types of human cancers[47], it may serve as a useful target for cancer therapy.

Here we demonstrate that the stability of VPRBP is controlled by USP2, a member of the USP family of deubiquitinases, and that knockdown or knockout of USP2 expression destabilizes VPRBP. Thus, small molecule inhibitors of USP2 effectively activate p53 function without disrupting the p53-Mdm2 interaction. Consistent with previous studies of USP2-knockout mice[49,50], USP2 inhibitors display no obvious toxicity in vivo. We further found that, in addition to repressing p53 function, the USP2-VBPBP axis also modulates the expression of PD-L1 (Supplementary Fig. 1). Indeed, inactivation of USP2 increases the levels of both p53 and PD-L1 in tumors, suggesting that the tumor suppression activity of USP2 inhibition may be potentiated by the PD-1/PD-L1 immune checkpoint blockade. Moreover, although a small-molecule inhibitor of USP2 alone can partially suppress the in vivo growth of p53-wild-type mammary tumor xenografts, more strikingly, USP2 inhibition and PD-1/PD-L1 blockade in combination promote vigorous tumor regression and long-term survival of all tumor-bearing mice. Thus, targeting the USP2/VPRBP pathway unleashes the latent tumor suppression activity of p53 in cancer cells, avoids the severe toxicities associated with Mdm2 inhibitors, and synergizes effectively with immune checkpoint blockade to achieve dramatic tumor regression in vivo.

## Results

### VPRBP inhibition activates p53 while also inducing p53-independent upregulation of PD-L1 expression

VPRBP was originally identified as a cellular protein that binds and modulates the transcriptional activity of HIV-1 viral protein R[51]. We and others have shown that VPRBP also acts as a transcriptional repressor that interacts with the C-terminal domain of p53 and antagonizes its transcriptional activity[45,47]. Like Mdm2, VPRBP is overexpressed in several tumor types and as such represents a potential target for reactivating p53 function in human cancer cells (Supplementary Fig. 2). As expected, RNAi-mediated depletion of VPRBP markedly elevates the

expression of p53 target genes (e.g., p21, TIGAR, PUMA, and Mdm2) in human osteosarcoma U2OS cells, but not in isogenic *p53*-null U2OS cells (Fig. 1a and Supplementary Fig. 3a). Interestingly, upon analysis of the expression profiles, we found that the levels of PD-L1 (also called *CD274*) were significantly induced upon VPRBP knockdown regardless of p53 status (Supplementary Fig. 3b). Indeed, as shown in Fig. 1b, siRNA-mediated depletion of VPRBP significantly increased both the mRNA and protein levels of PD-L1 in cell lines that do (human osteosarcoma U2OS and human melanoma A375) or do not (human lung carcinoma H1299), express wild-type p53, suggesting that VPRBP can regulate *PD-L1* expression regardless of p53 status. To validate whether VPRBP-mediated PD-L1 regulation is p53-independent, we performed a double knockdown of VPRBP and p53 in A375 cells. As shown in Fig. 1c, p53 knockdown had no effect on PD-L1 upregulation induced by VPRBP depletion. Moreover, VPRBP-mediated PD-L1 regulation was also observed in isogenic *p53*-null U2OS cells (Fig. 1d). Finally, upregulation of PD-L1 levels by VPRBP depletion was observed in a number of human cancer cell lines regardless of their p53 status (Fig. 1e). Together, these data indicate that inactivation of VPRBP enhances p53 function while also inducing PD-L1 expression, independent of p53.

### VPRBP acts as a repressor of IRF1-mediated transcriptional activation of the *PD-L1* gene

PD-L1 is widely expressed on tumor cells and multiple types of host cells including dendritic cells, macrophages and T cells in the tumor microenvironment (TME) and induced by cytokines such as IFNγ[52–56]. Numerous studies have shown that IRF1, a transcription factor that mediates IFNγ signaling, promotes both PD-L1 upregulation in tumor cells and tumor progression in vivo[53–58]. Interestingly, we identified VPRBP as a potential transcriptional corepressor of IRF1. As shown in Fig. 2a, Myc-tagged VPRBP was readily detected in the immunoprecipitated complexes of SFB-tagged IRF1. Conversely, Flag-tagged IRF1 was co-immunoprecipitated with SFB-tagged VPRBP (Fig. 2b). To ascertain whether VPRBP and IRF1 interact directly, we performed in vitro GST pull-down assays by incubating purified Flag-VPRBP with a GST-fusion protein containing full-length IRF1. As shown in Fig. 2c, VPRBP bound an immobilized GST-IRF1 fusion protein but not GST alone (Supplementary Fig. 4a). To evaluate this interaction under more physiological conditions, we performed co-immunoprecipitation assays with endogenous proteins from human lung carcinoma H1299 cells. As shown in Fig. 2d, endogenous IRF1 protein was co-precipitated by a VPRBP-specific antibody; conversely, endogenous VPRBP was co-precipitated by an IRF1-specific antibody in H1299 cells (Fig. 2e). The interaction of endogenous VPRBP and IRF1 was also validated in A549 and HEK293 cells (Supplementary Fig. 4b, c). Thus, IRF1 is a bona fide binding partner of VPRBP both in vitro and in vivo.

Next, we examined whether VPRBP modulates IRF1-dependent transcriptional activation of PD-L1. To this end, we co-transfected cells with expression vectors encoding either IRF1 alone, or IRF1 and VPRBP together, along with a luciferase reporter harboring the promoter sequences of PD-L1, which contain two IRF1 binding sites[57] (Fig. 2f). As expected, IRF1 expression strongly induced activation of the PD-L1 reporter (lane 3 vs. lane 1, Fig. 2g). However, co-expression of IRF1 with VPRBP led to a strong repression of the PD-L1 reporter (lane 4 vs. lane 3, Fig. 2g), suggesting that VPRBP is able to suppress the transcriptional activity of IRF1. We also examined whether VPRBP can modulate IRF1-mediated activation of the endogenous PD-L1 gene. Although VPRBP co-expression did not affect IRF1 protein levels, it significantly repressed IRF1-mediated induction of endogenous PD-L1 (Fig. 2h). In addition, we generated an *IRF1*-null H1299 cell line through CRISPR technology and validated that endogenous PD-L1 levels were reduced upon loss of IRF1 expression (Supplementary Fig. 4d). Moreover, while IFNγ treatment markedly stimulated the expression of both IRF1 and PD-L1 in native H1299 cells, IFNγ-mediated induction of PD-L1 was fully abrogated in *IRF1*-null H1299 cells (Supplementary Fig. 4d). To evaluate

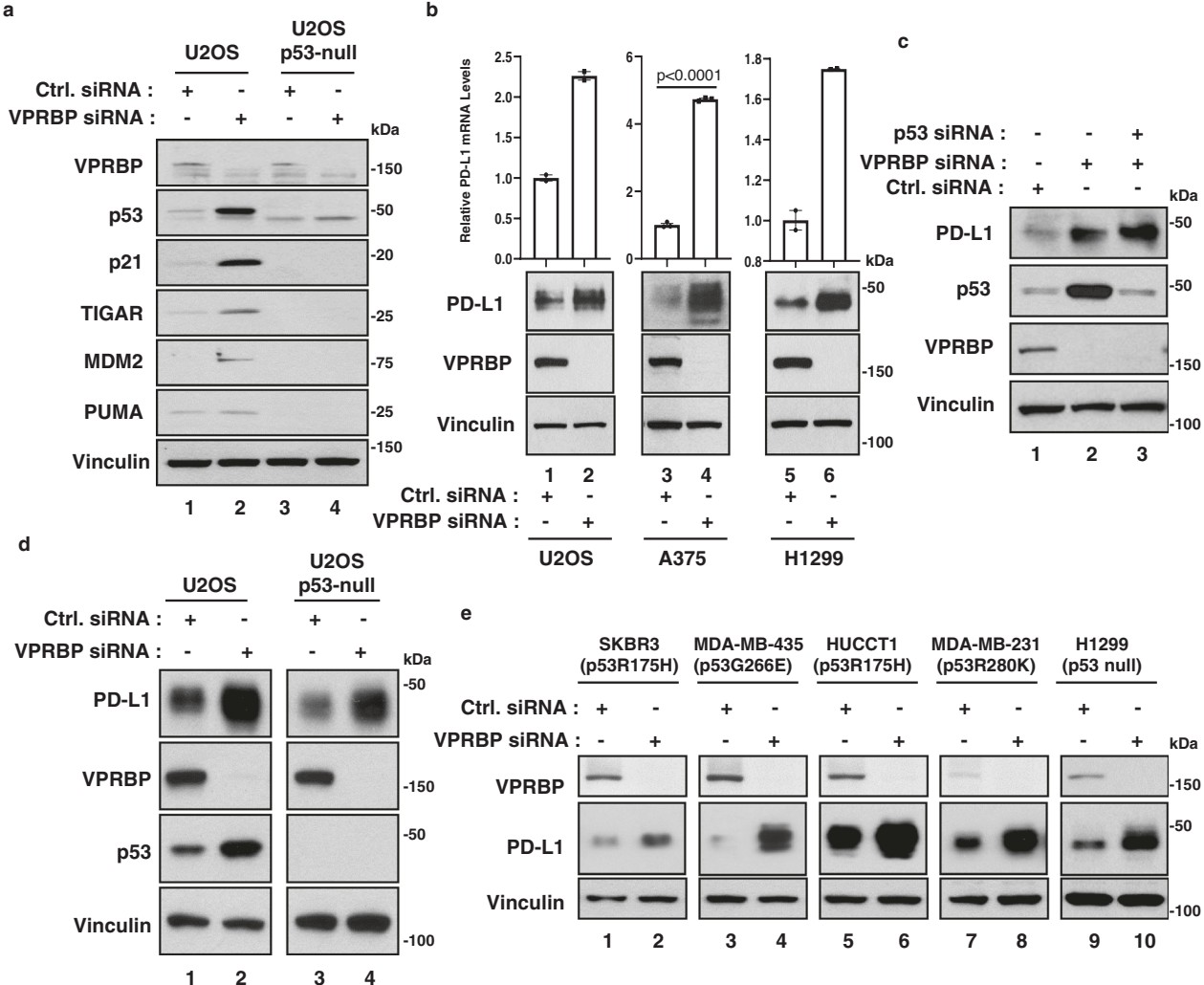

**Fig. 1 | Inactivation of VPRBP induces p53-independent upregulation of PD-L1 expression. a, d** Western blot analysis of U2OS crispr control or *p53*-null cells that were transfected with the control or VPRBP siRNA. **b** Representative qRT-PCR and western blot analysis of U2OS, A375, and H1299 cells that were transfected with control or VPRBP siRNA. *n*(U2OS and H1299) = 2 biologically independent samples, *n*(A375) = 3 biologically independent samples, mean ± SD, two-tailed unpaired *t*-test. **c** Western blot analysis of A375 cells that were transfected with control siRNA, VPRBP siRNA alone or VPRBP siRNA, and P53 siRNA. **e** Western blot analysis of the indicated human cancer cells that were transfected with control siRNA or VPRBP siRNA. All data are representative of two independent experiments. Source data are provided in the Source data file.

the role of endogenous VPRBP in modulating IRF1 function, we then examined whether IRF1-mediated activation of PD-L1 is affected by RNAi-mediated VPRBP depletion. As shown in Fig. 2i, the mRNA levels of endogenous PD-L1 were significantly upregulated by VPRBP depletion in both untreated and IFNγ-treated native H1299 cells, but not in isogenic *IRF1*-null cells (Fig. 2i). Moreover, a VPRBP mutant lacking the IRF1-binding domain (ΔAD, Fig. 2j and Supplementary Fig. 4e) failed to repress IRF1-dependent PD-L1 transactivation (Fig. 2k). These data demonstrate that VPRBP represses IRF1-mediated transactivation of the PD-L1 gene through its direct interaction with IRF1 (Supplementary Fig. 1).

### PD-L1 is degraded by VPRBP-induced ubiquitination

Although VPRBP plays an important role in transcriptional repression, VPRBP, is also called DCAF1 (DDB1−CUL4-associated-factor 1), that can function as a substrate recognition subunit of the CUL4-DDB1 ubiquitin E3 ligase complex[48]. Consistent with this notion, we found that VPRBP is present in both cytoplasmic and nuclear fractions as previously reported[51] (Supplementary Fig. 4f). As shown in Fig. 2i, a VPRBP mutant that fails to associate with the CUL4-DDB1 complex (ΔE3, Fig. 2j

and Supplementary Fig. 4g) still retains the ability to repress IRF1-dependent PD-L1 transactivation (Fig. 2l). Nevertheless, while VPRBP can clearly regulate PD-L1 gene expression independent of its associated E3 ligase activity, it is conceivable that VPRBP also influences PD-L1 levels through ubiquitin-mediated degradation. To explore this possibility, we used a two-step affinity chromatography protocol (anti-Flag M2−agarose beads and S-protein−agarose beads)[59,60] to isolate PD-L1-associated protein complexes from extracts of H1299 cells that stably express a C-terminal tagged PD-L1 (PD-L1-SFB) protein (Supplementary Fig. 5a). Analysis of the affinity-purified PD-L1-associated proteins by liquid chromatography mass spectrometry/mass spectrometry (LC−MS/MS) revealed four peptide sequences matching VPRBP (Supplementary Fig. 5b, c). To validate the interaction between VPRBP and PD-L1, we transfected H1299 cells with a Flag-tagged VPRBP expression vector in the presence or absence of a vector encoding SFB-tagged PD-L1. As shown in Fig. 3a, VPRBP was readily detected in the immunoprecipitated complexes of PD-L1. To ascertain whether VPRBP and PD-L1 interact directly, we performed in vitro GST pull-down assays by incubating purified Flag-tagged VPRBP with a GST-fusion protein containing full-length PD-L1. As shown in Fig. 3b, VPRBP

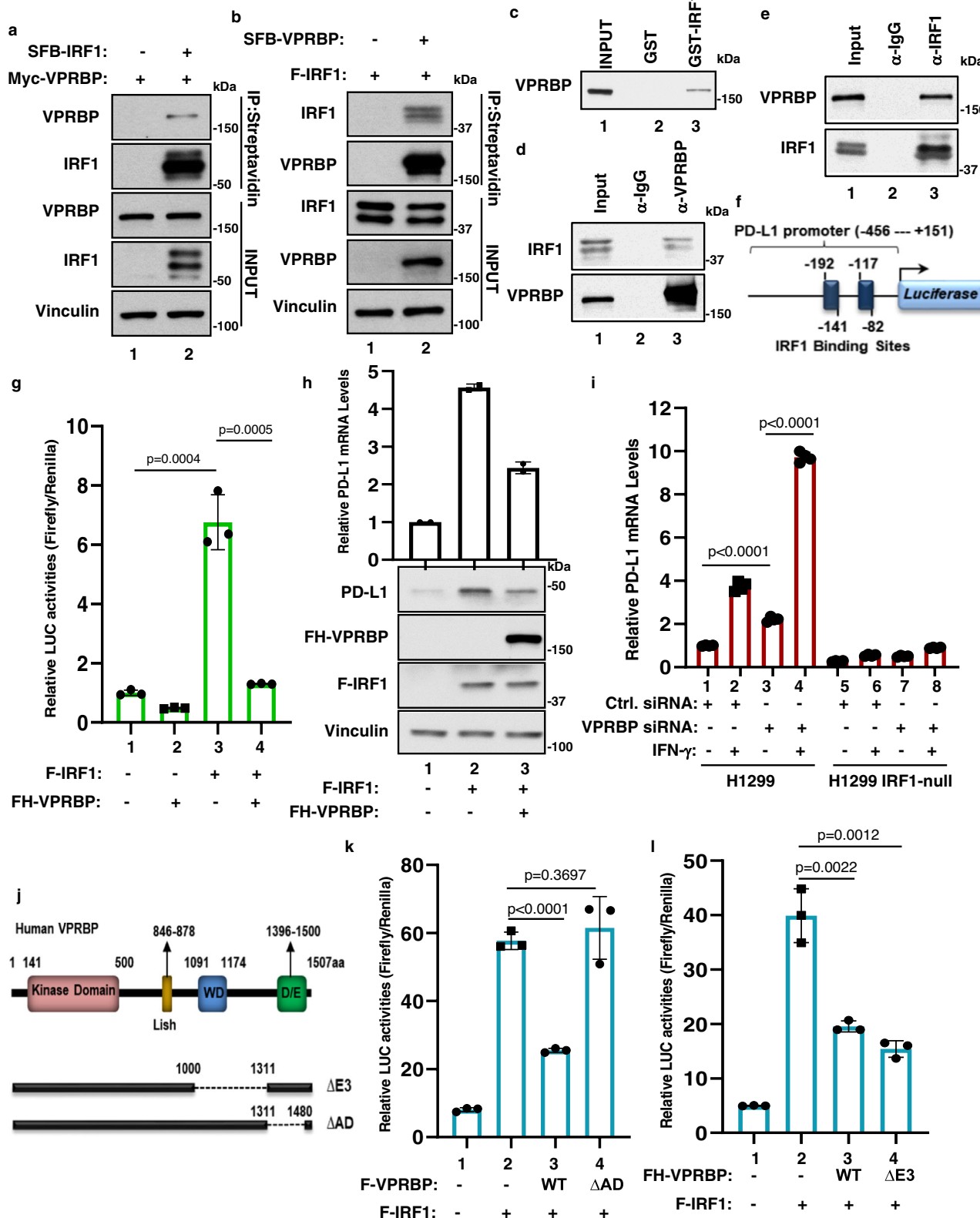

strongly bound the GST-PD-L1 fusion but not GST alone. More specifically, VPRBP bound a GST-fusion protein harboring the C-terminal intracellular domain (ICD, amino acids 231-290), but not the N-terminal (1-119) or middle (120-230) domains, of PD-L1 (Fig. 3c), while PD-L1 efficiently bound an N-terminal (NT), but not a mostly C-terminal (ΔNT), segment of VPRBP (Supplementary Fig. 5d, e). To evaluate this interaction under more physiological conditions, we performed co-

immunoprecipitation assays with endogenous proteins from native H1299 cells. As shown in Fig. 3d, the endogenous VPRBP protein was co-precipitated by the PD-L1-specific antibody but not by the IgG control antibody (upper panels); conversely, the endogenous PD-L1 protein was co-precipitated by the VPRBP-specific antibody but not by the IgG control (lower panels). These data demonstrate that VPRBP specifically interacts with PD-L1 both in vitro and in vivo.

**Fig. 2 | VPRBP acts as a repressor of IRF1-mediated transcriptional activation of PD-L1. a** Western blot analysis for VPRBP after immunoprecipitation (IP) of SFB-IRF1, with streptavidin beads, from H1299 cells transfected with Myc-VPRBP alone or with SFB-IRF1. **b** Western blot analysis for IRF1 after IP of SFB-VPRBP, with streptavidin beads, from H1299 cells transfected with F-IRF1 alone or with SFB-VPRBP. **c** Representative western blot analysis for pulldown of purified Flag-VPRBP with GST or GST-IRF1. **d** Western blot analysis for endogenous IRF1 after immunoprecipitation of endogenous VPRBP in H1299 cells. **e** Western blot analysis for endogenous VPRBP after immunoprecipitation of endogenous IRF1 in H1299 cells. **f** Schematic diagram representing the PDL1-luc construct that includes partial of PD-L1 promoter containing two IRF1 binding sites. **g** Luciferase activity of PDL1-luc in H1299 cells transfected with indicated constructs. $n = 3$ biologically independent samples, mean ± SD, two-tailed unpaired t-test. **h** Representative qRT-PCR and western blot analysis of H1299 cells transfected with empty vector, Flag-IRF1 alone or with Flag-HA-VPRBP. $n = 2$ biologically independent samples, mean ± SD. **i** qRT-PCR analysis of relative PD-L1 mRNA levels in H1299 parental and *IRF1*-null cells transfected with control siRNA or VPRBP siRNA for 48 h followed by 10 ng/ml of Interferon-gamma (IFN-γ) for additional 24 h. $n = 3$ biologically independent samples, mean ± SD, two-tailed unpaired t-test. **j** Schematic diagram of human VPRBP protein and deletion mutants. WD WD40 repeat domain, D/E aspartic acid (D) and glutamic acid (E)-rich domain. **k, l** Luciferase activity of PDL1-luc in H1299 cells transfected with indicated constructs. $n = 3$ biologically independent samples, mean ± SD, two-tailed unpaired t-test. All data are representative of at least two independent experiments. Source data are provided in the Source data file.

To understand the functional consequence of this interaction, we first examined whether VPRBP expression affects PD-L1 protein levels. Of note, PD-L1 levels were dramatically reduced upon co-expression of wild-type VPRBP (Fig. 3e), but not VPRBP-ΔNT (Fig. 3f), a mutant that binds poorly to PD-L1 (Supplementary Fig. 5d, e). Conversely, a PD-L1 mutant (PD-L1Δ60) that fails to bind VPRBP (Fig. 3c), was resistant to VPRBP-mediated degradation (Fig. 3g). These data suggest that a direct interaction between VPRBP and PD-L1 is required for the VPRBP-induced reduction in PD-L1 levels. Next, we examined whether VPRBP can induce PD-L1 ubiquitination in vivo. As shown in Fig. 3h, high levels of ubiquitinated PD-L1 were generated upon expression of wild-type VPRBP (lane 2 vs. lane 1), but not a mutant (VPRBP-ΔNT) that is deficient for PD-L1 binding (lane 3 vs. lanes 2). The ubiquitination levels of endogenous PD-L1 were increased upon VPRBP expression (Supplementary Fig. 5f). Moreover, the half-life of PD-L1 was significantly extended upon VPRBP knockdown (Fig. 3i, j). Since VPRBP acts as a substrate recognition subunit of a CUL4-DDB1 ubiquitin E3 ligase complex (CRL4$^{VPRBP}$), we examined whether other components of this complex also modulate PD-L1 stability. As shown in Fig. 3k, l, PD-L1 levels were upregulated upon knockdown of either DDB1 or Cul4A/B. In addition, VPRBP-mediated PD-L1 degradation was abrogated by a specific inhibitor (MLN4924) of the Cul4-E3 ligase, but not by a lysosomal inhibitor BafA1 (Supplementary Fig. 5g). the VPRBP-ΔE3 mutant, defective in interacting with the CUL4-DDB1 E3 ligase complex, exhibited impaired capability to degrade PD-L1 (Supplementary Fig. 5h). Collectively, these results demonstrate that VPRBP directly induces ubiquitin-mediated degradation of PD-L1 by acting as a substrate recognition subunit of the CRL4$^{VPRBP}$ E3 ligase complex (Supplementary Fig. 1). Thus, VPRBP can suppress PD-L1 protein levels through two distinct mechanisms: repression of IRF1-mediated transcription of the *PD-L1* gene (Fig. 2) and ubiquitin-mediated degradation of PD-L1 by the CRL4$^{VPRBP}$ E3 ligase (Fig. 3).

## VPRBP inhibition blocks tumor growth in vivo in a p53-dependent manner
To further elucidate the role of VPRBP in modulating p53 function, we tested whether inhibition of VPRBP expression affects tumor growth in immunodeficient nude mice. As expected, shRNA-mediated depletion of VPRBP activates p53 and induces the expression of p21 and PUMA in EMT6 mouse mammary tumor cells but not in isogenic *p53*-null EMT6 cells (Fig. 4a). Moreover, VPRBP depletion dramatically reduced the growth of mouse mammary xenograft tumors from native EMT6 cells (Fig. 4b, c), an effect that was largely abrogated in xenografts of isogenic *p53*-null EMT6 cells (Fig. 4b, d). These data demonstrate that VPRBP inhibition activates p53-mediated transcriptional activity and promotes p53-dependent tumor growth suppression in immunodeficient xenograft tumor models.

## Synergistic effects on tumor growth suppression by the combination of VPRBP inhibition and immune checkpoint blockade
Curiously, VPRBP inhibition potentially produces two opposing effects on tumor development by suppressing tumor cell growth by activation

of p53 (Fig. 4a–c) while also allowing tumor cells to evade immuno-surveillance through increased PD-L1 levels (Figs. 1–3). Thus, it may be possible to unleash the full therapeutic potential of VPRBP inhibition through immune checkpoint blockade. As expected, in addition to activating the p53 pathway, VPRBP depletion also markedly induced PD-L1 expression levels in both native and *p53*-null EMT6 cells (Fig. 4a), and similar results were obtained in *p53*-null 4T1 mouse mammary tumor cells (Supplementary Fig. 6a). These VPRBP-mediated effects on PD-L1 levels were further validated by FACS and qPCR analyses using four independent shRNAs against VPRBP (Supplementary Fig. 6b–d). Clinical studies have shown that the success of PD1-PD-L1 checkpoint blockade with either anti-PD1 or anti-PD-L1 antibody correlates positively with PD-L1 expression levels on the tumor cells[52,61–63]. Thus, we also examined the impact of VPRBP inhibition on the growth of EMT6 tumor xenografts in immunocompetent (i.e., Balb/c) mice. As shown in Fig. 4e, f, VPRBP depletion reduced tumor growth in a small subset of Balb/c mice (2 of 11 mice; Fig. 4e, panel II). This effect is much less pronounced than the uniform reduction of tumor growth observed in nude mice (6 of 6 mice; Fig. 4b), likely reflecting the ability of VPRBP depletion to upregulate PD-L1 expression in immunocompetent mice. If so, then PD1-PD-L1 checkpoint blockade may potentiate the tumor growth suppression activity of VPRBP depletion in these mice. As shown in Fig. 4e, intraperitoneal injections of anti-PD-1 monoclonal antibody reduced tumor growth in a subset of Balb/c mice (4 of 8 mice; Fig. 4e, panel III). Remarkably, VPRBP inhibition combined with anti-PD-1 treatment dramatically retarded tumor growth in all immuno-competent Balb/c mice tested (11 of 11 mice; Fig. 4e, panel IV). Moreover, a marked improvement in overall survival was observed in mice treated with VPRBP depletion and anti-PD-1 in combination relative to mice subjected to either treatment alone (Fig. 4f). The elevated levels of PD-L1 in EMT6 tumors with shVPRBP were validated by both the FACS analysis and immunohistochemistry (Supplementary Fig. 6e–i). Interestingly, although anti-PD-1 alone modestly affected the levels of tumor-infiltrating lymphocytes (TILs), the combination of anti-PD-1 treatment and VPRBP depletion induced a marked increase in TILs, including CD4$^+$, CD8$^+$/granzyme B$^+$ T cells (Fig. 4g, h). Immunohisto-chemically staining of CD8 of tumor sections from each treatment cohort exhibited similar effect (Supplementary Fig. 7a, b). Moreover, H&E staining and immunohistochemistry analysis of tumor proliferation marker Ki67 further confirmed that the combination of VPRBP silencing and PD-1/PD-L1 checkpoint blockade displayed much potent anti-proliferation effects (Supplementary Fig. 7c, d). These data demonstrate that by activating p53 function and triggering anti-tumor immunity, the combination of VPRBP inhibition and PD1-PD-L1 block-ade dramatically represses the growth of p53-wild-type tumors.

## USP2 is critical for controlling VPRBP stability
To further elucidate the mechanisms by which VPRBP modulates both p53 and PD-L1, we sought to identify potential regulators of VPRBP activity. Deubiquitinating enzymes (DUBs) often modulate the functions of specific E3 ligases through their ability to remove ubiquitin conjugates on the same protein substrates[64]. The human genome

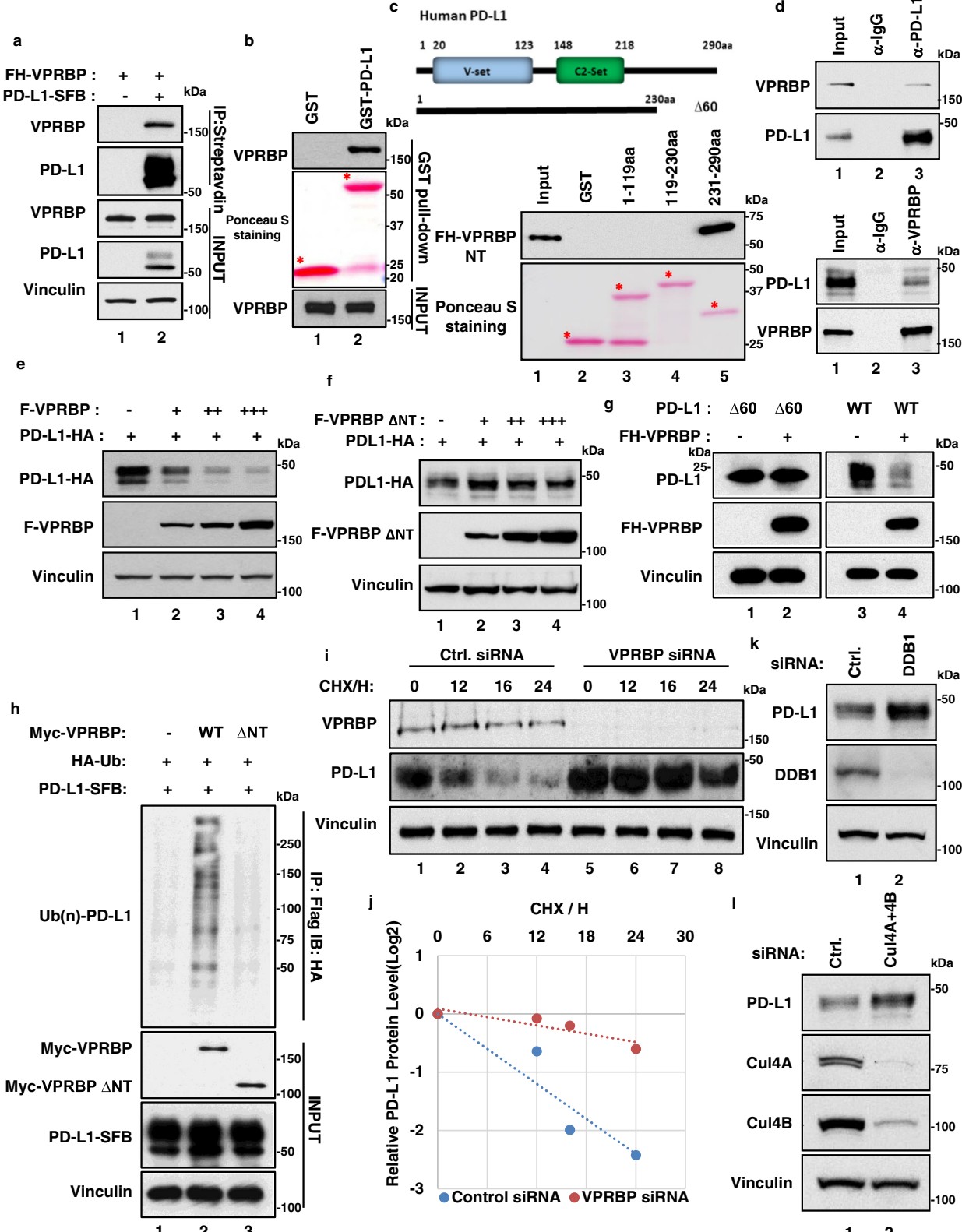

harbors ~100 DUB genes, of which about 60 belong to the ubiquitin-specific protease (USP) family of DUBs[65]. Interestingly, upon biochemical purification of protein complexes formed by the SFB-tagged-USP2 protein in 293 cells (Supplementary Fig. 8a), we identified USP2 as a potential binding partner of VPRBP by both mass spectrometry analysis (Supplementary Fig. 8b, c) and western blot analysis (Supplementary Fig. 8d). To further validate this interaction, we first

examined whether USP2 interacts with VPRBP in the cells through co-transfection. Western analysis revealed that Myc-tagged VPRBP co-immunoprecipitates with FLAG-HA-tagged USP2 (FH-USP2) in H1299 cells (Fig. 5a), while FH-USP2 co-immunoprecipitates with SFB-tagged VPRBP (Fig. 5b). Moreover, co-immunoprecipitation analysis of human melanoma A375 cell extracts indicate that endogenous VPRBP and USP2 proteins interact in vivo (Fig. 5c). This interaction appears to be

**Fig. 3 | PD-L1 is degraded by VPRBP-induced ubiquitination. a** Western blot analysis for VPRBP after immunoprecipitation of PD-L1-SFB, with streptavidin beads, from H1299 cells transfected with indicated constructs. **b** Western blot analysis for pulldown of purified Flag-VPRBP with purified GST-PD-L1. "*" indicates specific band. **c** Schematic representation of human PD-L1 protein and deletion mutant Δ60 (upper panel). V-set: Immunoglobulin V-set domain; C2-set: Immunoglobulin C2-set domain; Lower panel is western blot analysis for pulldown of purified Flag-HA-VPRBP NT with GST-PD-L1 deletion mutants "*" indicates specific band. **d** Western blot analysis for the interaction between endogenous VPRBP and PD-L1 in H1299 (upper panel) and MDA-MB-231 cells (lower panel). **e, f** Western blot analysis of H1299 cells transfected with PD-L1-HA alone, or with increasing amount of Flag-VPRBP (**e**) or ΔNT (**f**). **g** H1299 cells were transfected with PD-L1 wild type (WT) or deletion (Δ60) alone, or plus FH-VPRBP constructs. Whole cell extracts were subjected to western blot analysis. **h** HEK293T were transfected with PD-L1-SFB alone, or plus HA-Ub with or without Myc-VPRBP or ΔNT, after anti-Flag IP, immunoprecipitates were subjected to western blot analysis. **i, j** H1299 cells were transfected with control or VPRBP siRNA for 48 h followed by 50 μg/ml cycloheximide (CHX) treatment for indicated hours. Whole cell extracts were subjected to western blot analysis (**i**) and PD-L1 protein abundance was quantified with Image J software (**j**). **k, l** Western blot analysis of H1299 cells transfected with control siRNA, DDB1 siRNA (**k**) or Cul4A and 4B siRNA (**l**). All data are representative of two independent experiments. Source data are provided in the Source data file.

direct since purified Flag-VPRBP binds a purified GST-USP2 fusion protein but not GST alone (Fig. 5d). Of note, VPRBP protein levels were significantly increased upon co-expression with wild-type USP2, but not with the enzymatically-defective USP2-C276A mutant (Fig. 5e), suggesting that USP2 stabilization of VPRBP is mediated by its deubiquitinase activity. Indeed, expression of wild-type USP2, but not USP2-C276A, significantly reduced in vivo ubiquitylation of VPRBP (Fig. 5f). Moreover, RNAi-mediated depletion of endogenous USP2 reduced the levels of endogenous VPRBP protein (Fig. 5g), but not VPRBP mRNA (Fig. 5h), in both H1299 and Cal-33 cells. Endogenous VPRBP protein levels were also significantly decreased in H1299 cells by three independent USP2-specific siRNAs (Fig. 5i). To further validate the role of USP2 in regulating VPRBP stability, we used CRISPR/Cas9 technology to inactivate the USP2 gene in *p53*-null H1299 cells. Consistent with the above data, USP2 inactivation significantly reduced the levels of VPRBP proteins (Fig. 5j) and, importantly, the half-life of VPRBP was markedly decreased in *Usp2*-null cells (Fig. 5k, l). Collectively, these results indicate that USP2 binds, deubiquitinates, and thereby stabilizes VPRBP in vivo (Supplementary Fig. 1).

## Inhibition of USP2 activates p53 but Mdm2-mediated regulation of p53 is largely unaffected

Several small molecular inhibitors of USP2 enzymatic activity have been recently described, including ML364 and LCAHA[66,67]. Indeed, similar to USP2 depletion, ML364 treatment induced VPRBP destabilization and heightened PD-L1 protein levels in a variety of human tumor lines, including those that do (human lung carcinoma H460 and human melanoma A375 cells; Fig. 6a) or do not (H1299 cells; Fig. 6b) express wild-type p53. Similar results were also obtained using LCAHA, another small molecule inhibitor of USP2 (Fig. 6c). As expected, key transcriptional targets of p53 (e.g., p21, PUMA, and Mdm2) were activated upon ML364 treatment of tumor lines expressing wild-type p53 (H460 and A375) but not in *p53*-null H1299 cells (Fig. 6a, b). Conversely, the PD-L1 levels were induced upon USP2 inhibition in all cell types regardless of their p53 status.

Since Mdm2 levels were induced in p53-positive cells by USP2 inhibition (Fig. 6a, c), we examined whether USP2 inhibition affects Mdm2-mediated degradation of p53. As shown in Fig. 6d, although p53 levels were dramatically reduced by expression of exogenous Mdm2, treatment with the Mdm2 inhibitor RG7388 (also called idasanutlin) reversed this effect, confirming that RG7388 effectively blocks p53 degradation by Mdm2. In contrast, however, treatment with the USP2 inhibitor ML364 did not block Mdm2-mediated p53 degradation. Indeed, a functional p53-Mdm2 feedback loop is also retained by Crispr-derived *Usp2*-null cells, in which the Mdm2 half-life remains unaffected (Fig. 6e, also see Supplementary Fig. 9a); thus, p53 was readily degraded upon co-expression with exogenous Mdm2 (Fig. 6f) and endogenous Mdm2 levels were induced upon expression of p53 (Fig. 6g). To further support the notion that MDM2 retains its ability to effectively degrade p53 in the presence of USP2 inhibition, we performed additional experiments. To this end, we treated both control and Mdm2 knockdown cells with ML364 respectively. As expected, the levels of p53 were increased upon the treatment of ML364 but Mdm2

depletion by Mdm2 siRNA was able to further increase p53 levels in those cells, suggesting that Mdm2 is still functional in degrading p53 in the presence of the ML364 inhibitor (Supplementary Fig. 9b). Taken together, these data demonstrate that USP2 inhibition can activate p53 function and increase PD-L1 levels without significantly disrupting Mdm2-mediated regulation of p53.

## The combination of USP2 inhibition and the PD-1/PD-L1 immune checkpoint blockade completely suppresses tumor growth with no obvious toxicity

Our data demonstrate that USP2 inhibition activates p53 function by destabilizing VPRBP and that it does so without compromising p53-Mdm2 feedback loop. Since USP2 inhibition, unlike Mdm2 inhibition, does not cause severe damage to normal tissues, it may provide a clinically effective means of inducing the tumor suppression activity of p53. *Mdm2*-null mice suffer early embryonic lethality and cultured *Mdm2*-null cells are not viable unless p53 function is also inactivated[23,25,27,28,49,50]. In contrast, *Usp2*-null mice develop normally, displaying only mild phenotypic defects as adults, and *Usp2*-null cells from these animals are viable in culture[49,50]. Mdm2 inhibitors generally led to hematological toxicity, including thrombocytopenia and neutropenia, mainly caused by deregulated p53 with severely damaged bone marrows in cancer patients[10,15,22]. In contrast, upon the treatment with the USP2 inhibitor ML364, no obvious signs of hematologic disorder or microscopic tissue damage were observed (Fig. 7a, b and Supplementary Figs. 10 and 11a). No obvious damage was detected in bone marrows, spleens as well as other major organs upon the treatment (Fig. 7b and Supplementary Fig. 10). No discernable change in body weight was apparent in the ML364-treated mice (Supplementary Fig. 11b). As expected, the USP2 inhibitor ML364 induced p53 activation and expression of its target genes (e.g., p21, PUMA, and Mdm2) in EMT6 mouse mammary tumor cells but not in isogenic *p53*-null EMT6 cells (Fig. 7c). Since USP2-mediated stabilization of PD-L1 is p53-independent, the USP2 inhibitor ML364 readily induced PD-L1 levels in both native and isogenic *p53*-null EMT6 cells.

Since ML364 can activate the transcriptional functions of p53 without inducing toxicity in normal tissues, we next examined whether ML364 can also elicit the tumor suppression activity of p53. To evaluate the therapeutic potential of ML364 in a syngeneic mouse model, p53-wild-type EMT6 tumor cells were injected subcutaneously into the right flank of immunocompetent Balb/c mice and tumor growth was monitored every 2–3 days. After EMT6 inoculation, the mice were randomized into four treatment groups: vehicle-treated, ML364-treated alone, anti-PD-1-treated alone, and combination-treated. ML364-treated mice received a daily injection of ML364 (30 mg ML364 per kg body weight (mg/kg)) for 15 days while mAb-treated (anti-PD-1 or IgG isotype control) mice received four intraperitoneal injections (200 μg IgG mAb per mouse) spaced over ten days (Supplementary Fig. 11c). As expected, the levels of PD-L1 were elevated in tumors upon the ML364 treatment (supplementary Fig. 11d). While all mice in the vehicle-treated cohort died within 27 days of EMT6 inoculation, as might be expected, anti-PD-1 treatment displayed a therapeutic effect, conferring relatively long-term survival on approximately half the mice

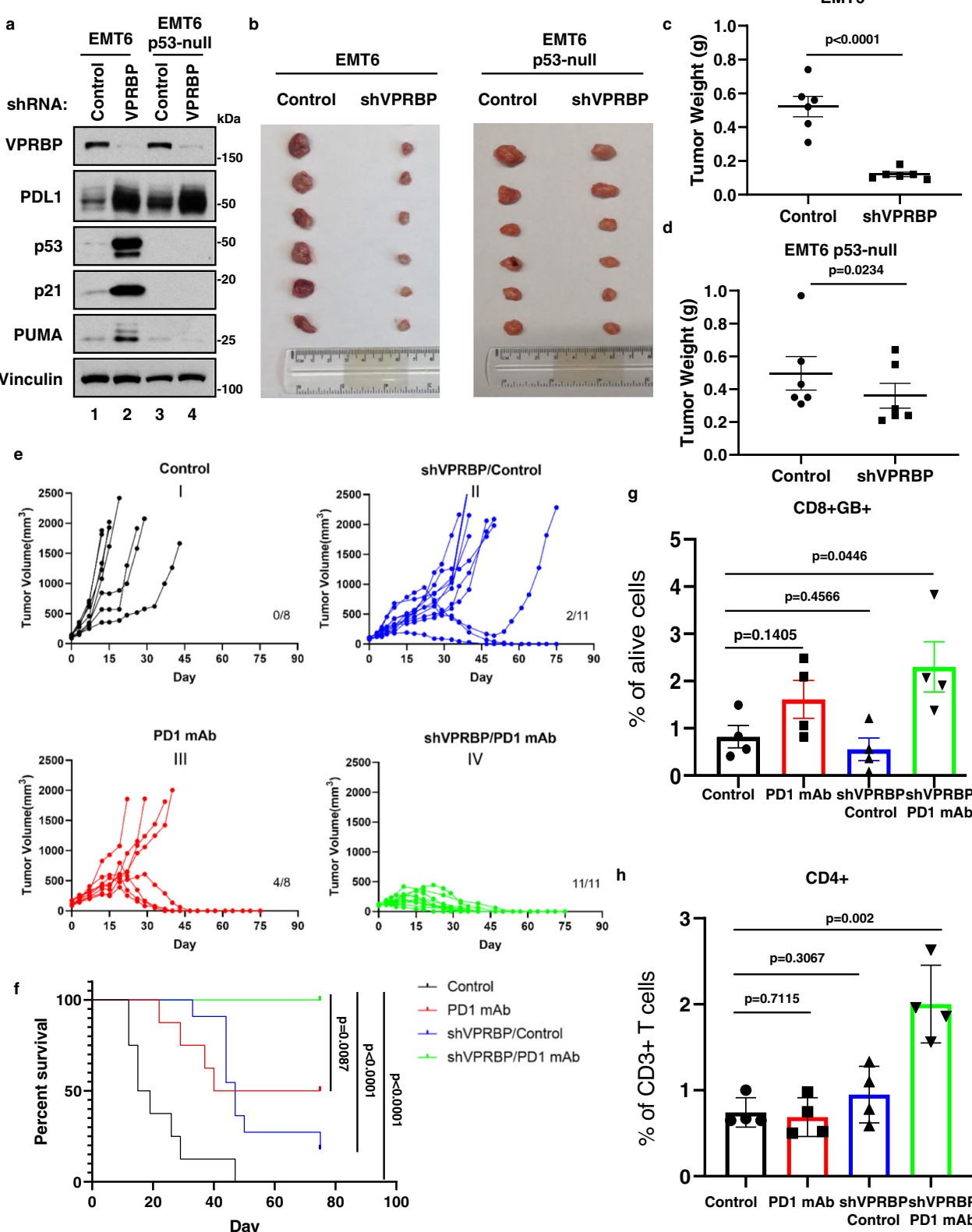

**Fig. 4 | Combination treatment with VPRBP inhibition and immune checkpoint blockade activates p53 and enhances immunotherapeutic efficacy. a** Western blot analysis of EMT6 and EMT6 *p53*-null cells transduced with or without shVPRBP lentiviruses. Data are representative of two independent experiments. **b** Images of tumors dissected from EMT6- and EMT6 *p53*-null-implanted nude mice. **c**, **d** represent the weight of EMT6 (**c**) and EMT6 *p53*-null (**d**) tumors. *n* = 6 tumors, mean ± SEM, two-tailed unpaired *t*-test. **e** Tumor growth curves of EMT6-implanted Balb/c mice treated with IgG isotype or anti-PD-1 monoclonal antibodies (PD1 mAb).

Cells were transduced with or without shVPRBP lentiviruses before inoculation. **f** Survival of EMT6-implanted Balb/c mice treated with IgG isotype or anti-PD-1 monoclonal antibodies (PD1 mAb). For Control and PD1 mAb, *n* = 8; for shVPRBP/ Control and shVPRBP/PD1 mAb, *n* = 11. Significance was determined by log-rank test. **g**, **h** Percentage of CD8[+]Granzyme B(GB)[+] cells (**g**) or CD4[+] cells (**h**) in EMT6 tumors analyzed by flow cytometry. *n* = 4 tumors, mean ± SD, two-tailed unpaired *t*-test. Source data are provided in the Source data file.

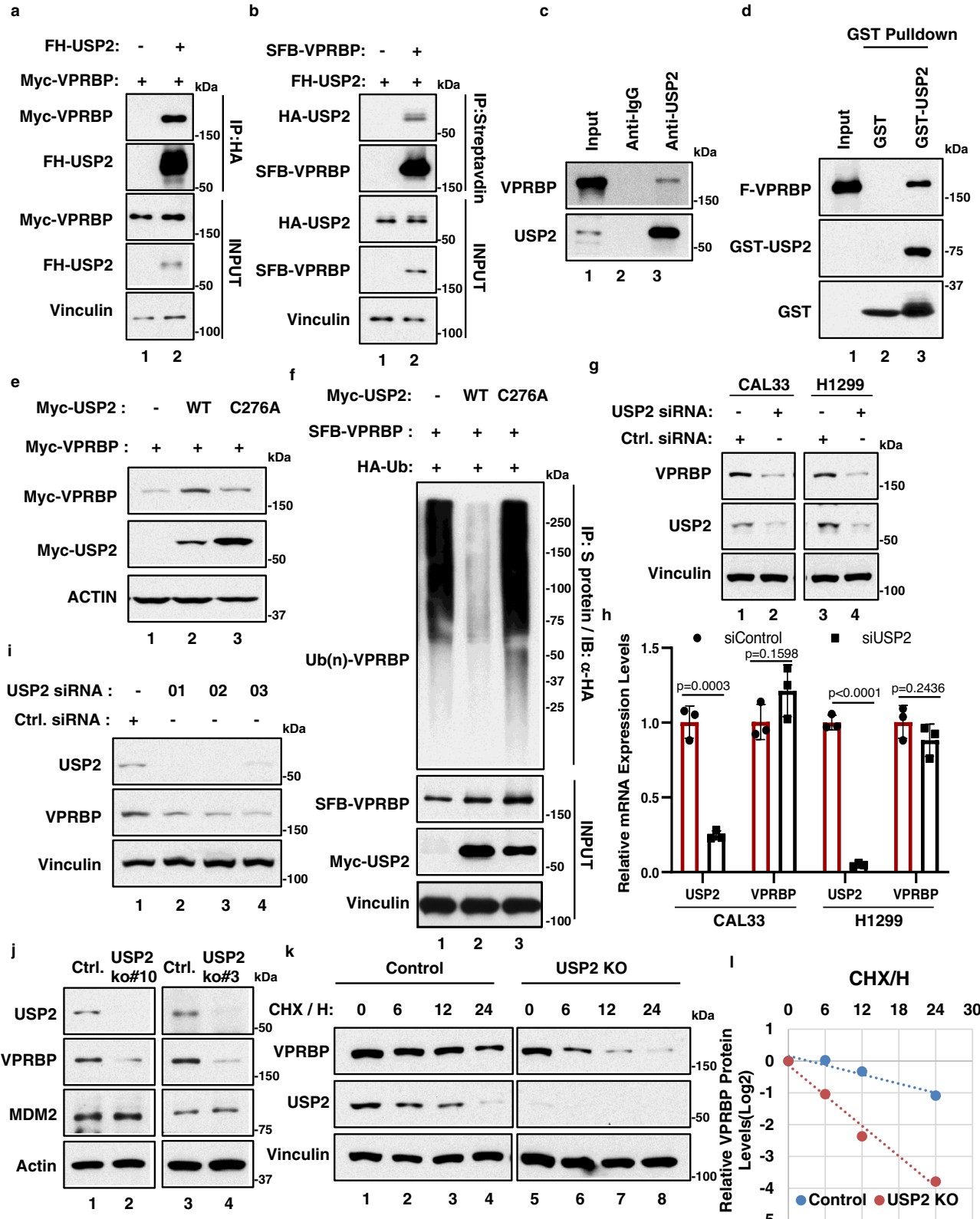

in the anti-PD-1-treated cohort (Fig. 7h). Of note, a similar proportion of the ML364-treated mice also achieved long-term survival, demonstrating that USP2 inhibition also exerts anti-tumor effects in vivo. Remarkably, all the mice of the combination-treated cohort (9 of 9) displayed dramatic tumor regression and long-term survival (Fig. 7e, g, h), indicating that the combination of USP2 inhibition and PD-1/PD-L1 blockade is an especially effective mode of tumor therapy.

Moreover, to further validate this notion, we performed similar experiments with other types of murine cancer cells. As shown in Fig. 7d, the USP2 inhibitor ML364 induced p53 activation and upregulation of PD-L1 in mouse melanoma cancer cell line B16F10 and mouse prostate cancer cell line RM-1. Indeed, synergistic effects induced by the combination of a small-molecule USP2 inhibitor and anti-PD1 monoclonal antibody were also readily reproduced in the RM-1 tumor-bearing

**Fig. 5 | VPRBP is a bona fide substrate of USP2 deubiquitinase. a** Western blot analysis for VPRBP after immunoprecipitation (IP) of FH-USP2, with HA beads, from H1299 cells transfected with Myc-VPRBP alone or with FH-USP2. **b** Western blot analysis for USP2 after immunoprecipitation (IP) of SFB-VPRBP, with streptavidin beads, from H1299 cells transfected with FH-USP2 alone or with SFB-VPRBP. **c** Western blot analysis for endogenous VPRBP after immunoprecipitation of endogenous USP2 and control IgG in A375 cells. **d** Western blot analysis of F-VPRBP pulled down by GST or GST-USP2 protein in in vitro GST-pull down assay. **e** Western blot analysis of Myc-VPRBP in H1299 whole cell extracts transfected with Myc-VPRBP alone, or plus Myc-USP2 wild-type (WT) or C276A. **f** Western blot analysis of ubiquitinated VPRBP (Ub(n)-VPRBP) after immunoprecipitation of SFB-VPRBP by S protein beads under denaturing condition in H1299 cells transfected with indicated

constructs. **g, h** Cal33 and H1299 cells were transfected with control or USP2 siRNA for 96 h. Whole cell lysates were subjected to SDS-PAGE followed by western blot analysis (**g**). Total RNA was extracted for cDNA synthesis and qPCR analysis (**h**). $n = 3$ biologically independent samples, mean ± SD, two-tailed unpaired $t$-test. **i** Western blot analysis of VPRBP in H1299 cells transfected with control or USP2 siRNA oligos. **j** Western blot analysis of VPRBP and MDM2 in H1299 control or USP2 knockout (KO) cells. **k, l** H1299 control and USP2 knockout (KO) cells were treated with 100 μg/ml cycloheximide (CHX) for indicated time. Whole cell extracts were subjected to SDS-PAGE followed by western blot analysis (**k**). VPRBP protein abundance was quantified with Image J software (**l**). All data are representative of at least two independent experiments. Source data are provided in the Source data file.

C57BL/6 mice without any obvious toxicity (Supplementary Fig. 12a–e). In addition, our data showed that both CD4 + and CD8 + TILs were significantly increased in tumors with the combination treatment (Fig. 4g, h and supplementary Fig. 7a, b). Moreover, we performed CD4 and CD8 depletion assays to further elucidate which TILs play the pivotal role in tumor regression. To this end, we injected 200 μg/ml of mouse CD4- or CD8-specific antibodies to mice one day before the combination therapy (Supplementary Fig. 13a). FACS analysis of the mouse splenocytes demonstrated that CD4 or CD8 T cells were completely depleted (Supplementary Figs. 13b and 14). As expected, the combination treatment of ML364 and PD-1 antibodies effectively repressed tumor growth ((Supplementary Fig. 13c, d). Interestingly, upon the deletion of either CD4 or CD8 cells in the same treatment by ML364 and PD1mAb on those mice, loss of CD8 cells completely abrogated the effect of tumor growth suppression whereas depletion of CD4 cells failed to show any obvious effect. These data demonstrate that CD8 + T cells, but not CD4 + T cells, are the primary effector cells underlying the combination treatment in these tumor models.

Finally, to ascertain whether USP2 inhibition potentiates the efficacy of PD-1/PD-L1 blockade by enhancing p53 function in a tumor cell autonomous manner, we also examined the effects of combination therapy in Balb/c mice bearing *p53*-null EMT6 tumors. As shown in Fig. 7f, the combination therapy was much less potent on *p53*-null EMT6 tumors, in which it conferred long-term survival on only a subset of mice, similar to the effect of anti-PD-1-only treatment on p53-wild-type EMT6 tumors. These data indicate that a combination of USP2 inhibition and PD-1/PD-L1 blockade potently suppresses tumor growth in vivo, and that the contribution of USP2 inhibition to this effect is dependent on the tumor cell autonomous activity of p53.

## Discussion

The p53 protein was dubbed the "guardian of the genome" because of its crucial role in coordinating cellular responses to genotoxic stress[68,69]. In this capacity, p53 suppresses tumor formation by promoting outcomes, such as apoptosis or cell cycle arrest, that limit the propagation of cells with damaged or unstable genomes. In addition to genotoxic stress, p53 is also triggered by a variety of other stresses that may arise in tumor cells, such as hypoxia, nutrient deprivation, oxidative stress and oncogene activation[3,4,70]. Although the various outcomes of p53 activation are beneficial for the management of stressed cells, as well as limiting tumor formation, most of them are potentially harmful to the physiology of normal (i.e., unstressed) cells. Thus, it is critically important to restrain p53 function in unstressed cells and to downregulate p53 in stressed cells once the stress has been properly managed. Given the wide variety of stresses that can potentially activate p53 function, cells need a similarly diverse array of mechanisms to restrain p53 function. Indeed, a number of p53 regulators have been identified, many of which serve as both co-repressors of p53 transcriptional activity and E3 ubiquitin ligases that modify the stability and/or subcellular localization of p53, such as Mdm2, SET, Sirt1, ARF-BP1/Mule, Mdmx, Cop-1, and Pirh2[23,24,33]. The Mdm2 protein is especially important in this regard as it maintains p53 at low levels in normal

cells by targeting it for proteasomal degradation. Moreover, since the *Mdm2* gene is itself a transcriptional target of p53, the two proteins form a sensitive autoregulatory loop that keeps p53 levels low in normal cells, while allowing rapid activation of p53 function when these cells experience stress[25,34].

Although p53 is the most commonly mutated driver of human cancer, a majority of human tumors still retain wild-type p53 genes/functions[1]. Therefore, reactivating the latent tumor suppression function of p53 for therapeutic purposes should be beneficial for many cancer patients[71–74]. To this end, a variety of Mdm2 inhibitors that block the Mdm2/p53 interaction have been developed, several of which readily reactivate p53 function and suppress the growth of cultured tumor cells[10]. Unfortunately, although these Mdm2 inhibitors are highly effective against human tumors that retain wild-type p53, they also induce serious damage in normal tissues by allowing deregulated p53 activation in normal cells[10,12,15,22]. These results are not entirely unexpected as numerous studies have documented the toxic effects of Mdm2 inhibition on normal development and cell homeostasis[27,28]. For example, *Mdm2*-null mice suffer early embryonic lethality and *Mdm2*-null cells do not survive in culture[75,76]. Since co-inactivation of p53 restores normal development in *Mdm2*-null mice and allows *Mdm2*-null cells to survive in culture, the toxic effects of Mdm2 loss are likely a consequence of deregulated p53 function. It is very likely that disrupting Mdm2-mediated regulation of p53 by Mdm2 inhibitors may be responsible for the toxic effects on normal tissues[10,22,33].

In this study, we identified the USP2-VPRBP axis as a pathway by which p53 function is regulated in tumor cells. Like MDM2, VPRBP is overexpressed in human cancers and suppresses p53 function by repressing its transcriptional activity[45,47]. Moreover, VPRBP, also called DCAF1 (DDB1–CUL4-associated-factor 1), can also function as a substrate recognition subunit of the CUL4-DDB1 ubiquitin E3 ligase complex to promote ubiquitin-dependent degradation of p53[48]. Here we show that VPRBP levels are tightly controlled by USP2, a deubiquitinase that binds and stabilizes VPRBP, and that loss of USP2 function can activate p53 responses, including its tumor suppressive effects. Moreover, we demonstrate that the in vivo growth of mammary tumor xenografts in mice can be suppressed in a p53-dependent manner by inactivation of either USP2 or VPRBP.

Notably, unlike mice lacking a functional *Mdm2* gene, *Usp2*-null mice appear to develop normally and display no obvious signs of toxicity in their normal tissues[49,50]. Moreover, by using both USP2 inhibitors and *Usp2*-null cells, we found that the levels of endogenous Mdm2 can be activated in a p53-dependent manner upon USP2 inhibition (Fig. 6a–c, g) and Mdm2-mediated regulation of p53 is largely unaffected (Fig. 6d, f). In accord with these observations, the treatment of tumor-bearing mice with a small molecule USP2 inhibitor induces p53-mediated tumor regression with no apparent damage to normal tissues (Supplementary Fig. 10). The precise mechanism by which USP2 inhibition is not toxic compared to Mdm2 inhibition in vivo remains to be further elucidated. Although p53 can be activated robustly under both conditions, in contrast to Mdm2 inhibition, USP2 inhibition neither affects Mdm2 induction by p53 nor suppresses

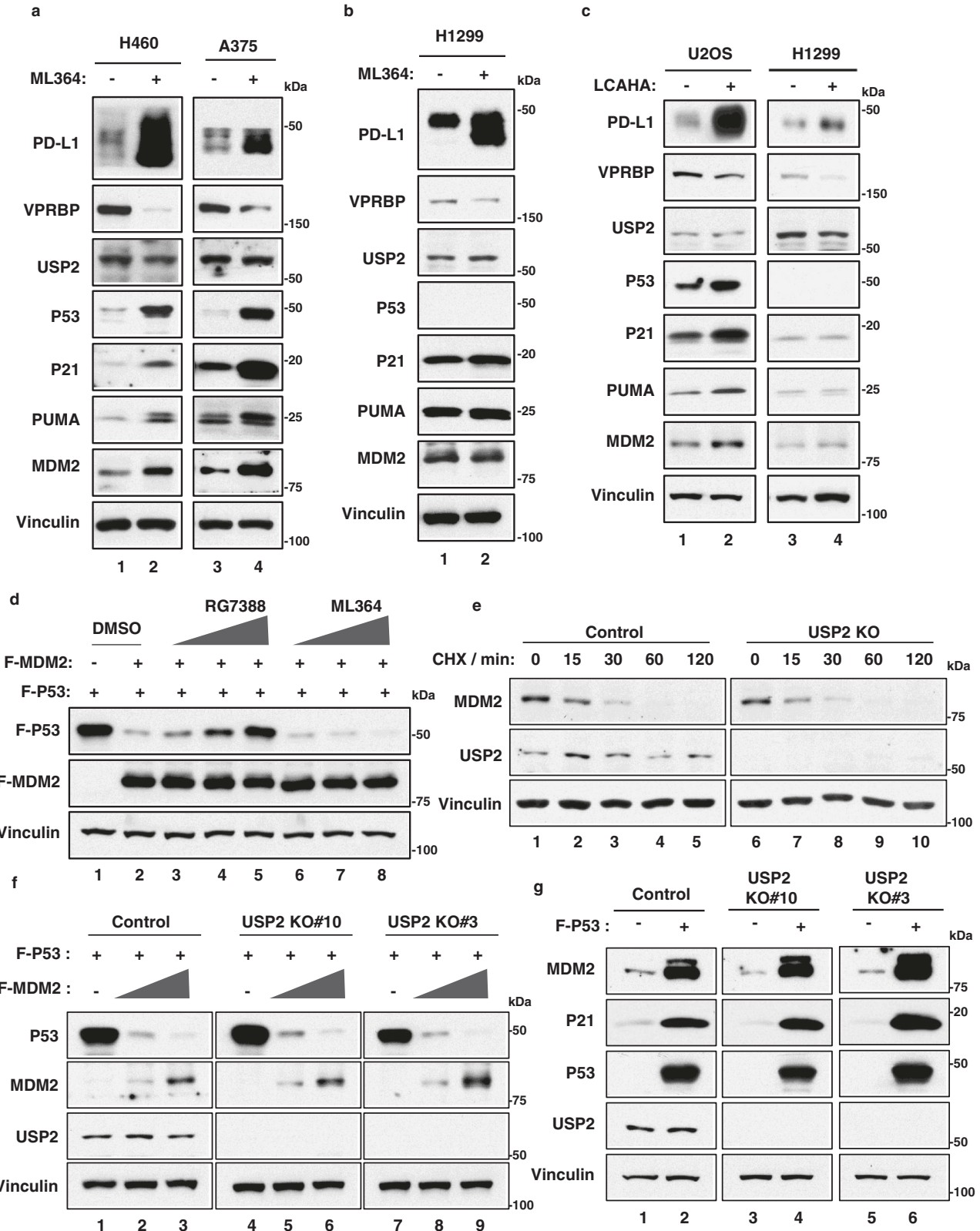

Mdm2-mediated repression of p53. It is very likely that USP2 inhibition does not significantly affect the negative p53-Mdm2 feedback loop. Previous studies indicate that activation of p53 alone does not automatically result in severe toxicity as observed in the "Super-p53" mice[28,29] but the negative p53-Mdm2 feedback loop is critical for normal cell homeostasis[27,31]. Thus, keeping the negative p53-Mdm2 feedback loop intact is likely the major factor to circumvent the toxicity

issue upon USP2 inhibition. Moreover, numerous studies indicate that USP2 is able to regulate other cellular factors potentially involved in tumorigenesis[65]. Future studies are clearly needed to examine whether p53-independent functions of USP2 may also contribute to the toxicity level as well as the effect in tumor growth suppression.

Clinical studies have shown that the therapeutic efficacy of PD1-PD-L1 checkpoint blockade correlates with PD-L1 expression levels on

**Fig. 6 | Inactivation of USP2 activates p53 and increases the PD-L1 levels but p53-Mdm2 feedback loop is fully intact. a, b** H460, A375 (**a**), and H1299 (**b**) cells were treated with 20 μM ML364 for 72 h. Whole cell extracts were subjected to SDS-PAGE followed by western blot analysis. **c** U2OS and H1299 cells were treated with 20 μM LCAHA for 72 h. Whole cell extracts were subjected to SDS-PAGE followed by western blot analysis. **d** H1299 cells were pretreated with DMSO, 1 μM, 2.5 μM, and 10 μM RG7388 or ML364 for 30 min and then transfected with F-P53 alone, or plus F-MDM2 for 24 h. Whole cell extracts were subjected to SDS-PAGE followed by

western blot analysis. **e** H1299 control and USP2 knockout (KO) cells were treated with 100 μg/ml cycloheximide (CHX) for indicated time. Whole cell extracts were subjected to SDS-PAGE followed by western blot analysis. **f** Western blot analysis in H1299 control or USP2 knockout (KO) cells transfected with F-P53 alone or plus increasing amounts of F-MDM2 constructs. **g** Western blot analysis of endogenous P21 and MDM2 expression in H1299 control or USP2 knockout (KO) cells transfected with or without F-P53 constructs. All data are representative of two independent experiments. Source data are provided in the Source data file.

the tumor cells[61–63,77]. Thus, while high PD-L1 levels prevent cytotoxic T cells from targeting tumor cells, they can also serve as a selective marker to stratify patients for PD1-PD-L1 checkpoint blockade therapy. IFNγ signaling strongly induces PD-L1 expression on tumor cells through IRF1-mediated transcriptional activation of the *PD-L1* gene. Interestingly, we found that the USP2-VPRBP axis modulates PD-L1 expression in a p53-independent manner through two distinct mechanisms (Supplementary Fig. 1). On the one hand, VPRBP acts as transcriptional corepressor that binds IRF1 and thereby inhibits expression of the *PD-L1* gene. On the other hand, VPRBP also acts as a substrate recognition subunit of the CUL4-DDB1 ubiquitin E3 ligase complex and that directly binds PD-L1 and targets it for ubiquitin-mediated degradation. As both mechanisms serve to reduce PD-L1 levels (Supplementary Fig. 1), the USP2-VPRBP axis should suppress PD-L1-mediated immune evasion and thereby enhance tumor immunosurveillance. Thus, our study indicate that VPRBP inhibition produces two seemingly opposing effects on tumor development by suppressing tumor cell growth by activation of p53 but also allowing tumor cells to evade immunosurveillance through increased PD-L1 levels. Indeed, USP2 inhibition alone partially induces tumor growth suppression but more strikingly, the combination of USP2 inhibition and PD-1/PD-L1 blockade promote vigorous tumor regression and long-term survival of all tumor-bearing mice. Interestingly, clinical studies demonstrated that the success of PD1/PD-L1 blockade by either anti-PD1 or anti-PD-L1 antibody has a positive correlation with PD-L1 expression levels in tumor cells[62,77–79]. Thus, although high levels of PD-L1 prevents cytotoxic T cells from effectively targeting tumor cells, it apparently also serves as a potential selective marker for patient stratification for PD1–PD-L1 blockade therapy. It will be very interesting to examine whether this combination treatment is able to reactivate both p53 function and the immune response in the certain tumors that failed to respond to the immunotherapy. Several studies indicate that the levels and functions of PD-L1 are tightly regulated by other cellular factors through different mechanisms including glycosylation, ubiquitination, deubiquitination, and acetylation[77,80–82]. Nevertheless, the USP2-VPRBP axis is able to target both PD-L1 and p53 simultaneously. Thus, our study not only provides a new layer of PD-L1 regulation but also has significant implications for the treatment of the tumors that retain wild type p53 to improve the therapeutic efficacy.

A recent study showed that approximately 65% of human tumors retain wild-type p53 function[1]. Thus, induction of its latent tumor suppression activity may be a potential therapeutic option for a broad spectrum of human cancers. Our results show that the tumor suppression activity of p53 can be unleashed by inhibition of the USP2/VPRBP pathway, and that this mode of p53 activation should circumvent the toxicities that have arisen in clinical trials of Mdm2 inhibitors. Future studies are clearly warranted to examine whether the combination of USP2 inhibitors and PD1-PD-L1 checkpoint blockade is especially effective in the treatment of human cancer patients in clinical trials.

## Methods
### Mice
All mouse experiments were approved by the Institutional Animal Care and Use Committee (IACUC) at Columbia University Health Sciences Center under the supervision of the Institute of Comparative Medicine.

All mice were bred in a pathogen-free facility with a 12 h light/dark cycle at 20° ± 3 and 40–50% humidity. Murine tumor cells were suspended in 50 μl of DMEM and 50 μl of Matrigel matrix (corning, Cat#354248), then injected subcutaneously into 6–8-weeks-old mice (Nude mice: RRID:IMSR_CRL:088;Balb/c:RRID:IMSR JAX:000651;C57 BL/6:RRID:IMSR JAX:000664). For nude mice, tumors were dissected and weighed two weeks after inoculation. For Balb/c and C57BL6 mice, tumor sizes were measured with caliper 2–3 times per week and tumor volumes were calculated with the formula $L \times W^2 \times 0.5$. Maximum tumor volume doesn't exceed 2000 mm³. For bioluminescence imaging, mice were injected intraperitoneally with 3 mg of D-luciferin (PerkinElmer, Cat# 122799) and photographed with IVIS(In Vivo Imaging Systems) Spectrum Optical Imaging System (PerkinElmer).

In all, $2.5 \times 10^5$ of EMT6-Luc cells that stably expressed luciferase reporter gene mixed with Matrigel matrix were subcutaneously injected into the right flank of female Balb/c mice. At 3 day after cell injection, mice were treated with either vehicle or 30 mg/kg ML364 by daily intraperitoneal injections for 15 days. At 5 day after cell injection, each cohort (vehicle- and ML364-treated mice) were randomized into two groups which were treated with 200 μg of IgG isotype or α-PD-1 mAb by intraperitoneal injection, two-three times per week for total four times. Treatment plan was shown as Supplementary fig. 11c. Tumor sizes measurement and IVIS imaging were performed as described above. Mouse was sacrificed once tumor volume reaches 1000 mm³. For CD4/8 depletion assay, mice were treated with 200 μg of IgG isotype or α-CD4/8 mAb by intraperitoneal injection. Treatment plan was shown as Supplementary Fig. 13a.

In all, $2.5 \times 10^5$ of RM-1 cells mixed with Matrigel matrix were subcutaneously injected into the right flank of male C57BL/6 mice. At 5 day after cell injection, mice were randomly divided into 4 groups treated with Vehicle, ML364, α-PD-1 mAb alone or combination therapy. Treatment plan was shown as Supplementary Fig. 12a. Tumor sizes measurement was performed as described above. Mouse was sacrificed once tumor volume reaches 2000 mm³.

For the toxicity assay, female Balb/c mice were randomized into two groups and injected intraperitoneally with vehicle or 30 mg/kg ML364 daily. At day 10 after treatment, blood samples were collected from the submandibular veins for complete blood count in Columbia University Irving Medical Center Diagnostic Lab Services Core. After then mice were euthanized and main organs were collected and fixed with 10% formalin for 24 h. Fixed tissues were then sent to Columbia University Irving Medical Center Histology Core for the preparation of paraffin sections for hematoxylin and eosin (H&E) staining and immunohistochemistry anlysis.

### Cell culture, transfection, and lentivirus transduction
Following cell lines were used in this study: EMT6 (ATCC Cat# CRL-2755),293 T (ATCC Cat# CRL-3216), H1299 (ATCC Cat# CRL-5803), U2OS ATCC Cat# HTB-96), CAL33 (Creative Bioarray Cat# CSC-C0479), HUCCT1 (Creative Bioarray Cat# CSC-C9200W), SKBR3 (ATCC HTB-30), A549 (ATCC Cat# CCL-185), A375 (ATCC Cat# CRL-1619),H460 (ATCC Cat# HTB-177), SKBR3 (ATCC Cat# HTB-30), MDA-MB-435 (ATCC Cat# HTB-129), MDA-MB-231 (ATCC Cat# HTB-26), and RM-1 (ATCC Cat# CRL-3310).

Cells were routinely maintained in DMEM media supplemented with 10%FBS, 100 units/ml penicillin and 100 μg/ml streptomycin in

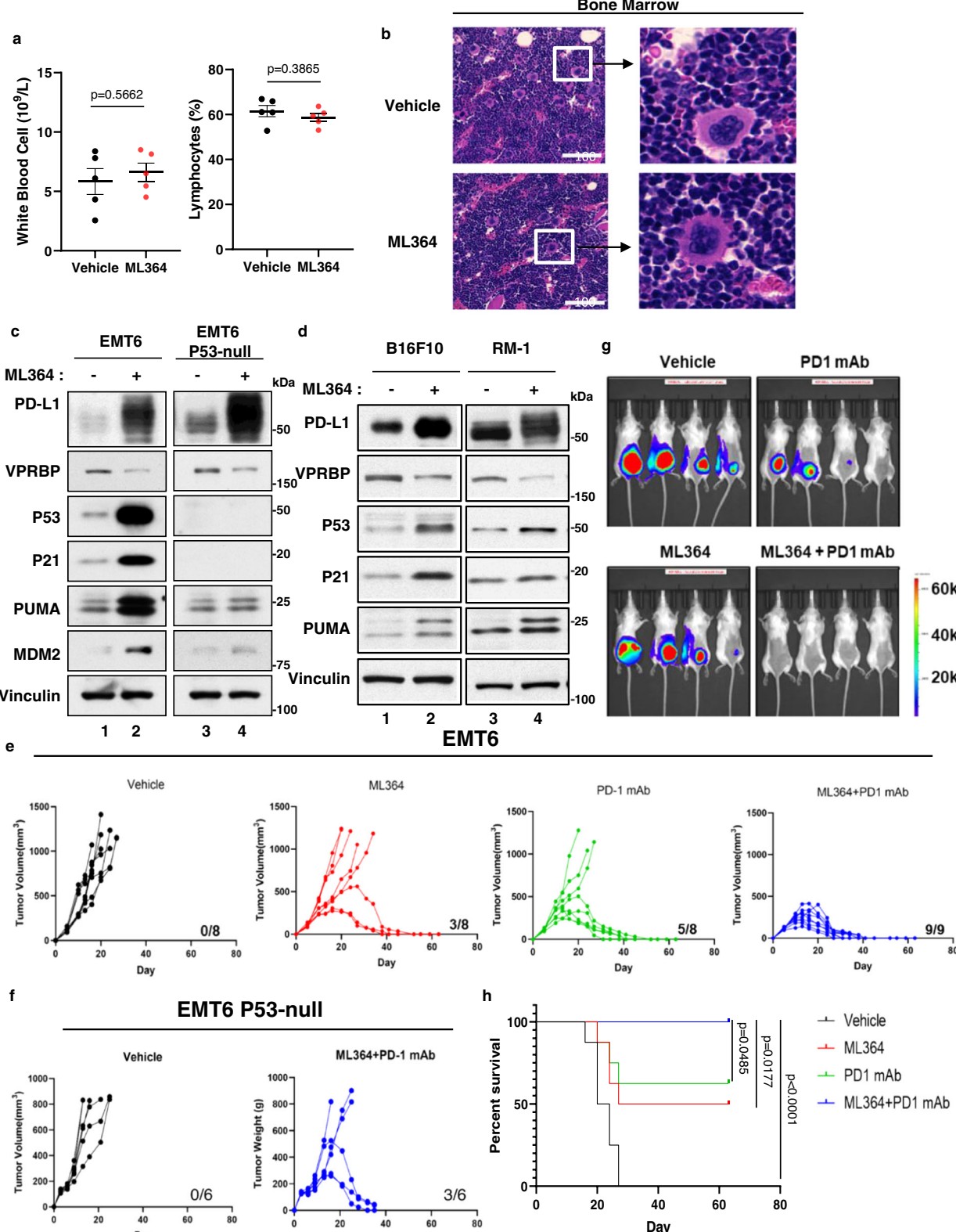

**Fig. 7 | Combination of USP2 inhibition and PD1/PD-L1 blockade leads to complete tumor regression with no obvious toxicity. a** Complete blood count of Balb/c mice treated with vehicle or 30 mg/kg ML364 for 10 days. $n = 5$ mice, mean ± SEM, two-tailed unpaired $t$-test. **b** Hematoxylin and Eosin (H&E) staining in bone marrow of Balb/c mice treated with vehicle or 30 mg/kg ML364 for 10 days. $n = 3$ images per group. **c** EMT6 and EMT6 *p53*-null cells were treated with 20 μM ML364 for 72 h. Whole cell extracts were subjected to SDS-PAGE followed by western blot analysis. Data are representative of two independent experiments.

**d** Western blot analysis of B16F10 and RM-1 cells treated with 20 μm ML364 for 72 h. Data are representative of two independent experiments. **e, f** Tumor growth curves of EMT6 (**e**) or EMT6 *p53*-null (**f**) cells-implanted Balb/c mice treated with indicated therapies. **g** Bioluminescence images of EMT6-impanted Balb/c mice treated with indicated therapies. **h** Survival of EMT6-implanted Balb/c mice treated with indicated therapies in **e**. Significance was determined by log-rank test. Source data are provided in the Source data file.

incubator at 37 °C with 5% CO2. All cell lines were not authenticated and were tested negative for mycoplasma contamination. Transfection of constructs and siRNA oligos was performed with lipofectamine 3000 reagents (Thermofisher scientific, Cat#L3000150) as manufacture's user guide. For lentivirus packaging, HEK293T cells were transfected with shVPRBP-pLKO, Δ8.9 and pCMV-VSVG constructs. At 48 h post-transfection, media containing viruses was harvested and filtered through 0.45 μm syringe filter. Before adding to cells, viruses were concentrated by using Lenti-X Concentrator (Clotech, Cat# 631321) as manufacturer's manual and resuspended with complete growth media. EMT6 cells were transduced with viruses overnight and kept growing for 3 days to get shVPRBP pool cells. The VPRBP knockdown efficiency was determined by QPCR or western blot analysis.

## shRNA and siRNA
VPRBP siRNA smartpool (Dharmacon, Cat# L-021119-01-0005); p53 siRNA smartpool (Dharmacon, Cat# L-003329-00-0005); Control siRNA smartpool (Dharmacon, Cat# D-001810-10-50); CUL4A siRNA smartpool (Dharmacon, Cat# L-012610-00-0005); CUL4B siRNA smartpool (Dharmacon, Cat# L-017965-00-0005); DDB1 siRNA smartpool (Dharmacon, Cat# L-012890-00-0005); USP2 siRNA#1-3 (Qiagen, Cat# 1027416_5; Cat# 1027416_6; Cat# 1027416_8) VPRBP shRNA#1-4 (Milliporesigma, Cat# TRCN0000265223; Cat# TRCN0000265224; Cat# TRCN0000251843; Cat# TRCN0000251844).

## Antibodies
Following primary antibodies were used for co-IP assay and western blot analysis: anti-USP2 (1:1000 dilution, Abgent Cat# AP2131c, RRID:AB_2212429); anti-PD-L1 (1:1000 dilution, Cell Signaling Technology Cat# 29122, RRID:AB_2798970); anti-PD-L1 (1:1000 dilution, Cell Signaling Technology Cat# 13684, RRID:AB_2687655); anti-VPRBP (1:2000 dilution, Bethyl Cat# A301-888A, RRID:AB_1524107); anti-VPRBP (1:1000 dilution, Santacruz Biotechnology Cat# sc-376850, RRID:AB_2905506); anti-mouse PD-L1 (1:1000 dilution, Abcam Cat# ab213480, RRID:AB_2773715); anti-IRF1 (1:1000 dilution, Cell Signaling Technology Cat# 8478, RRID:AB_10949108); anti-CUL4A (1:1000 dilution, Cell Signaling Technology Cat# 2699, RRID:AB_2086563); anti-CUL4B (1:1000 dilution, Sigma-Aldrich Cat# HPA011880, RRID:AB_1847340); anti-HA(1:5000 dilution, Roche Cat# 11867431001, RRID:AB_390919): anti-IRF1 (1:1000 dilution, Santa Cruz Biotechnology Cat# sc-74530, RRID:AB_2126826); anti-p53 (1:1000 dilution, Santa Cruz Biotechnology Cat# sc-126, RRID:AB_628082); anti-mouse p53(1:1000 dilution, Leica Biosystems Cat# NCL-L-p53-CM5p, RRID:AB_2895247); anti-PUMA (1:1000 dilution, Santa Cruz Biotechnology Cat# sc-28226, RRID:AB_2064827); anti-p21 1:250 dilution, (Santa Cruz Biotechnology Cat# sc-53870, RRID:AB_785026); anti-Actin (1:5000 dilution, Sigma-Aldrich Cat# A5441, RRID:AB_476744); anti-Flag (1:2000 dilution, Sigma-Aldrich Cat# F3165, RRID:AB_259529); anti-vinculin(1:5000 dilution, Sigma-Aldrich Cat# V9131, RRID:AB_477629). Following second antibodies were used for western blot: Peroxidase AffiniPure Goat Anti-Mouse IgG (1:5000 dilution, Jackson Immunoresearch Cat# 115-035-146, RRID:AB_2307392) and Peroxidase AffiniPure Goat Anti-Rabbit IgG (1:5000 dilution, Jackson Immunoresearch Cat# 111-035-045, RRID:AB_2337938). IgG isotype and monoclonal antibody used in mouse models are as following: IgG isotype (200 ng/mouse, Bio X Cell Cat# BE0089, RRID: AB_1107769); anti-mPD-1 (200 ng/mouse, Bio X Cell Cat# BE0273, RRID: AB_2687796); IgG isotype (200 ng/mouse, Bio X Cell Cat# BE0090, RRID:AB_1107780); mCD4 (200ug/mouse, Bio X Cell Cat#0003, RRID:AB_1107642) and mCD8 (200ug/mouse, Bio X Cell Cat#0061, RRID:AB_1125541). Antibodies used for FACS analysis are as following: PE-PD-L1 (1:20 dilution, BioLegend Cat# 124308, RRID: AB_2073556); BV421-CD8a (1:20 dilution, BioLegend Cat# 100737, RRID: AB_10897101); BV421-CD4 (1:20 dilution, BioLegend Cat# 100437, RRID: AB_10900241) and APC-

Granzyme B (1:20 dilution, BioLegend Cat# 372204, RRID:AB_2687028).

## Western blot and co-immunoprecipitation analysis
Whole cell extracts were prepared with Flag lysis buffer (50 mM Tris-HCl,pH 8.0, 137 mM NaCl, 1 mM NaF, 1 mM NaVO3, 1% Triton X-100, 0.2% sarkosyl, 0.5 mM DTT, 0.5 mM PMSF and 10% glycerol) containing fresh-added protease inhibitors. For cytosolic and nuclear fractions, the cell pellet was firstly incubated with Harvest buffer (10 mM Hepes (pH 8.0), 50 mM NaCl, 0.5 M sucrose, 0.1 mM EDTA, and 0.25% Triton X-100) containing fresh-added protease inhibitors for 5 min on ice followed by centrifugation @ 120 g for 10 min. The supernatant was cytosolic fraction. After wash twice with buffer A (10 mM Hepes (pH 8.0), 10 mM KCl, 0.1 mM EDTA, and 0.1 mM EGTA), nuclear pellet was lyzed with BC100 (50 mM Tris-HCl, pH 8.0, 100 mM NaCl, 0.2% Triton X-100, and 10% glycerol) containing fresh-added protease inhibitors for 30 min on ice to get the nuclear fraction. Protein concentration was measured with the protein assay dye reagent (Bio-Rad, Cat# 5000006) as manufacturer's user guide. 20-60μg total proteins were loaded to and separated in SDS-PAGE precast gels, then transferred to nitrocellulose membrane. After incubation with primary antibodies for overnight at 4 °C, HRP-conjugated secondary antibodies were used and western blot signals were detected on autoradiographic films after incubating with ECL substrate.

For co-immunoprecipitation assay, cells were lyzed and mild sonicated in BC100 buffer containing fresh-added proteinase inhibitors. Whole cell extracts were incubated with anti-Flag M2 beads (Sigma, Cat# A2220), S protein agarose beads (Sigma, Cat# 69704) or streptavidin sepharose™ beads (GE healthcare, Cat# 17511301) overnight at 4 °C. The next day, after wash with BC100 buffer, immunoprecipitates were eluted with flag peptide (Sigma, Cat# F3290), glycine-HCL, pH2.5 or 2 mg/ml biotin (Sigma, Cat# B4501), pH 8.0 respectively. Ubiquitination assays were performed under denaturing condition. Briefly, whole cell extracts were supplemented with SDS (final concentration: 1%) and boiled for 5 min to denature proteins. After that, whole cell extracts were 1:10 diluted with cell lysis buffer to lower SDS concentration to 0.1%. Agarose beads were then added for immunoprecipitation.

## GST-pulldown assay
GST-fused proteins and F-VPRBP proteins were expressed and purified in *E.coli* Rosetta (DE3) competent cells (Milliporesigma, Cat#70954-4) and HEK293T cells respectively. After incubation of GST or GST-fused protein and F-VPRBP with GST resin overnight at 4 °C, beads were washed 5 times with BC100 buffer, then boiled with SDS loading buffer. Precipitates were subjected to western blot analysis and Ponceau S staining.

## Quantitative real-time RT-PCR analysis and primers
Total RNA was isolated using TRIZOL reagent (Thermofisher scientific, Cat#15596018) and reversely transcribed using Superscript IV VILO master mix (Thermofisher scientific, Cat# 11756050) according to manufacturer's protocol. Quantitative PCR was performed in triplicates with Power SYBR Green PCR master mix (Thermofisher scientific, Cat# 4368708) and 7500 Fast Real-Time PCR system (Applied Biosystems). QPCR primers are listed as below: GAPDH F: ACACCAT GGGGAAGGTGAAG, GAPDH R: AAGGGGTCATTGATGGCAAC; VPRBP F: CAGGGTGCACTTCTGAGTGAT, VPRBP R: GCAAGGCCATGCAGGT AT; PD-L1 F: GGACAAGCAGTGACCATCAA, PD-L1 R: GTG TGCTGGTCA CATTGAAAA; USP2 F: AGAACGGGAAGACAGTAGGA, USP2 R: CGAA-GACCGTAGAACAGTAACC; mouse GAPDH F: AACAGCAACTCCCACT CTTC, mouse GAPDH R: CCTGTTGCTGTAGCCGTATT; mouse PD-L1 F: GATCCATCCTGTTGTTCCTCAT, mouse PD-L1 R: CGCCACATTTCTCC ACATCTA; mouse P53 F: GCCATGGCCATCTACAAGAA, mouse P53 R: AATTTCCTTCCACCCGGATAAG.

## Luciferase assay

pGL3-PDL1-Luc reporter constructs were generated by inserting the partial PDL1 promoter region (−456 to +151) to pGL3-Luc empty vector. pGL3-PDL1-Luc reporter and Renilla control reporter were co-transfected with or without F-IRF1 alone, FH-VPRBP alone or F-IRF1 and FH-VPRBP in H1299 cells for 24 h. The relative luciferase activity was determined in triplicates with Dual Luciferase Reporter Assay System (Promega, Cat# E1910) as manufacturer's user guide.

## Establishment of P53 or USP2 knockout cells by Crispr technology

EMT6 cells were transfected with pCW-CAS9 and pLKO empty vector or pLKO-P53 crispr gRNA by using Lipofectamine 3000. pLKO empty vector and pCW-CAS9 constructs are gifts from Dr. Laura Pasqualucci. pLKO-P53 crispr gRNA were generated by inserting P53 crispr gRNA (target sequence: ACCATCGGAGCAGCGCTCA) into pLKO empty vector. 1μg/ml of doxycycline (Sigma, Cat#D9891) was used to induce CAS9 expression. At 48 h post-transfection, cells were selected with 2 μg/ml puromycin (Sigma, Cat# P9620) for 3 days to get the control and *p53*-null pools, which were then used to grow single clones. Single clones were firstly determined by anti-P53 (cm-5) western blot. Next, potential *p53*-null clones were treated with MDM2 antagonist RG7388 (Medchem Express, Cat# HY-15676) to stabilize P53 followed by western blot to determine the protein level of p53 and its downstream targets such as P21 and PUMA. Clones that have undetectable p53 and no change of p53 downstream targets upon RG7388 treatment were defined as *p53*-null clones. Finally, *p53*-null clones were further identified by genomic DNA sequencing to confirm the DNA editing by using P53 specific primers (F: TGATCGTTACTCGGCTTGTC; R: GTCTGCCTG TCTTCCAGATAC).

H1299 control and USP2 knockout cells were generated by using similar method with control double nickase plasmid (Santacruz biotechnology, sc-437281) and USP2 double nickase plasmid (Santacruz biotechnology, sc-411243-NIC). USP2 knockout clones were first identified with anti-USP2 antibody by western blot analysis and further confirmed by genomic DNA sequencing using USP2 specific primers (F: GAGTCTTTGAATGGCCAG GA; R: CTGTCCAGCTTCTGGGTTAG).

## Single-cell suspension generation from tumor tissue and flow cytometry analysis

Tumor tissues were dissected and minced, then digested with 1 mg/ml Collagenase (Thermofisher Scientific, Cat# 17018-019) in DMEM for 1 h at 37 °C. Cells were filtered through 70 μm cell strainer, then incubated with RBC Lysis buffer (Sigma, Cat# 11814389001) for 3 min to remove red blood cells. In all, $1 \times 10^6$ cells were stained with LIVE/DEAD fixable far red dead cell stain kit (Thermofisher Scientific, Cat# L10120) followed by control Igg isotype or primary anitbodies staining for 30 min on ice in dark. FACS analysis was performed with BD FACSCalibur.

## Reporting summary

Further information on research design is available in the Nature Portfolio Reporting Summary linked to this article.

## Data availability

The mass spectrometry data generated in this study have been deposited in the PRIDE (Proteomics IDEntifications Database) under accession code PXD040473 and PXD040477. The remaining data are available within the article and Supplementary Information. Source data are provided with this paper.

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

## Acknowledgements

This work was supported by the National Cancer Institute of the National Institutes of Health under Award R35CA253059, RO1CA258390 and R01CA254970 to W.G. This work was also supported by National Institutes of Health grant RO1CA227450 to R.B. We acknowledge the support from the Herbert Irving Comprehensive Cancer Center (HICCC; P30 CA13696) and thank the Molecular Pathology and Proteomics of Shared Resources of HICCC. The content is solely the responsibility of the authors and does not necessarily represent the official views of the National Institutes of Health.

## Author contributions

Conception and experimental design: J.Y. and W.G. Methodology and data acquisition: J.Y., O.T., D.W., and H.L. Analysis and interpretation of data: J.Y., O.T., and W.G. Manuscript writing: J.Y., R.B., and W.G.

## Competing interests

The authors declare no competing interests.
