## [Peer Review File · Nature Communications]

Reviewers' Comments:

Reviewer #1:

Remarks to the Author:

This is a very interesting and comprehensive study. The authors identified a new regulatory mechanism of USP2/VPRBP/p53/PD-L1 axis and suggested USP2 as a promising target to induce tumor suppression. It could lead to the development of therapeutic strategies for cancers.

There are a few things that can be done to enhance the feasibility and reliability:

1. Line 185. The analysis of the expression profiles including PD-L1 was not shown in the manuscript.
2. In Figures 4 and 7, PD-L1 IHC staining is required to demonstrate its upregulation in vivo in tumors (potentially in peripheral tissues as well to demonstrate safety despite enhanced expression in the periphery)
3. In the discussion sections, the authors need to discuss or speculate the specific mechanism of the synergy of PD-L1 blockade and USP2 inhibition and its specificity to the tumor environment.
4. In the discussion sections, the authors need to discuss how their new finding link to or distinguish from several other studies that demonstrate regulatory mechanisms of PD-L1 stability by post-translational modifications (i.e. glycosylation, ubiquitination, deubiquitination, and so on).
5. Figure number seems incorrect in line 511 "Supplementary Fig.8"
6. Recommend reading through for grammar mistakes to enhance current readability.

Reviewer #2:

Remarks to the Author:

The manuscript by Yi and coworkers describes an approach of treating tumors by combining the targeting of VPRBP/USP2 axis and PD1/PDL1 immune checkpoint. This group previously identified VPRBP/DCAF1 as a p53 binding protein and inhibitor. The current work further expanded on this connection and presented several new findings: (1) Identifying VPRBP/USP2 interaction and regulation. (2) Showing USP2 and VPRBP repress the activity of IRF1 to regulate PDL1 expression. (3) VPRBP also ubiquitinates PDL1 to regulate its level. (4) Combined depletion of VPRBP and anti PD1 mab treatment is effective therapy for tumor. (5) Small molecule inhibitor of USP2 also activates p53 and synergize with anti PD1 mab in tumor treatment to prolong survival.

Overall the manuscript presented a substantial body of data that are quite clear on the physical and functional interactions of the various proteins, and the anti-tumor effects of the USP2/anti PD1 combination therapy are strong. Therefore the findings have potential clinical implications given the possibility of developing USP2 small molecule inhibitor drugs.

A potential issue the authors need to address is the over interpretation of their results. In addition to reporting the findings, the authors made a strong statement that drugging VPRBP/USP2 is an alternative way of activating p53 without affecting the MDM2 feedback loop, thus is superior to MDM2 inhibitors that are known to have toxicities.

There are published studies suggesting that knockout of VPRBP/DCAF1 or USP2 activate p53 through affecting MDM2, possibly because DCAF1 regulates ribosomal biogenesis that is linked to MDM2 regulation (PMID: 33355139). Therefore, the strong p53 activation after targeting VPRBP and USP2 are likely caused by disruption of the MDM2 feedback loop. The authors' own result showing MDM2 overexpression still degrades p53 after USP2 inactivation does not mean that the MDM2 loop is fully functional.

It is also premature to conclude that USP2 inhibitor is superior to MDM2 inhibitors, since they were not directly compared in mice due to lack of potent mouse MDM2 inhibitors. It is true that USP2 inhibitors may have advantages due to effects on other targets besides p53. One could argue that a lower dose of MDM2 inhibitor will be non toxic but can synergize with anti PD1 therapy even better since it does not cause undesirable increase of PDL1 expression. In fact such synergistic activity has been reported (PMID: 33504557).

Therefore, in my opinion the title of manuscript and some conclusions and model (Fig.S13) are somewhat misleading and overtly strong.

Other specific comments:

Fig1d legend is missing.

Fig3h. The ubiquitination assay figure labels and legend are confusing. The figure was labeled "Denaturing", PDL1-SFB was pulled down by FLAG IP, which will not work under denaturing condition. Similar label also appears in Fig5f. Fig5g.

The definition of an intact MDM2 feedback loop is not clear and lacks experimental support. Presumably it means that p53 is able to induce MDM2 expression and the MDM2 is fully capable of inhibiting p53. Showing MDM2 overexpression being able to degrade p53 in the absence of USP2 is not sufficient for the conclusion (Fig6g).

In Fig.4a, VPRBP knockdown induced p53 strongly while also inducing MDM2. This effect is quite similar to DNA damage or ribosomal stress where MDM2 is functionally suppressed despite being induced by p53. This is consistent with previous reports that VPRBP disruption affects ribosomal biogenesis, which then affects MDM2 activity.

The authors stated in their introduction that VPRBP regulates p53 through MDM2-independent mechanism (line 153). This is not a fair representation of the literature, given the report from Yue Xiong lab (PMID: 33355139). Previous work also described USP2 inhibition leads to p53 activation through decreasing MDM2 expression (PMID: 27351221).

Reviewer #3:

Remarks to the Author:

In this manuscript, the authors demonstrate a dual role of USP2-VPRBP axis is another upstream pathway regulating protein stability of p53, and that USP2 inhibition retains the intact of MDM2-p53 pathway can potentially be used as an alternative, less toxic strategy to target wild-type p53 in tumors. Additionally, the author also showed that the USP2-VPRBP pathway regulate PD-L1 independently of p53, thereby sensitizing tumor response to anti-PD-1 therapies. The discovery is quite interesting and potentially impactful. However, the manuscript suffered from changing cell lines for different experiments/assays, and it is unclear if the proposed mechanism is indeed universal for all the models being used in this study. The authors should clarify if the effect is indeed universal. It is uncertain how USP2-VPRBP affects tumor growth. Does its inhibition on PD-L1 counteracts the tumor-suppressive effect mediated by p53? Further experiments are needed to clarify the notion. A few additional concerns about the manuscript are listed below:

Major point:

1. Are other TFs known to regulate PD-L1 such as Myc involved in the effect of VPRBP on PD-L1?
2. Fig. 2D-2E: It is puzzling why the interaction was demonstrated in A549 and 293 cells while the regulation of VPRBP in PD-L1 are unclear.
3. Fig. 3: To demonstrate the importance of VPRBP-CUL4-DPP1 in PD-L1 stability, it is essential to demonstrate if the $\Delta E3$ mutant of VPRBP affects PD-L1 protein stability.
4. To gain further insight into the relative contribution of immune system in the antitumor effect mediated by VPRBP KD, it would be informative to compare the antitumor tumor effects on immunodeficient and syngeneic tumor models.
5. CD4 and CD8 depletion assays are needed to strengthen which (or both) is/are the primary effector cells underlying the effect of VPRBP KD and anti-PD-1 combination.
6. The authors showed that VPRBP has a dual effect. One is to stabilize p53 for tumor suppression; the other is to induce PD-L1 expression that presumably suppresses T cell activity. It is unclear what is the primary mechanism underlying the effect of VPRBP in the syngeneic tumors? The authors need to reconcile these two potentially opposing mechanisms. It is puzzling why VPRBP KD

does not affect CD4 or CD8 T cell populations given its effect on PD-L1 expression. Does VPRBP KD affect activities of CD4 and/or CD8 T cells?

7. It would be important to compare the effects of MDM2i and USP2i on endogenous p53 expression in p53-WT cells, particularly in those EMT6 and RM1 cells, used for tumor growth assays.

8. It would be critical to compare the effects of $\Delta E3$ and ΔAD mutants of VPRBP KD on tumor growth and the response to anti-PD-1 therapy.

Minor point:

1. The manuscript focuses on the dual effects of USP2-VPRBP pathway. However, the title did not fully represent the body of the work. A title amendment is suggested.

#RE: NCOMMS-22-29249

Authors Response to Reviewer #1:

We greatly appreciate the positive view on our study by Reviewer #1. We have taken all the issues raised from the reviewer #1 very seriously. In this revised manuscript, we have provided the new data as requested and also made the changes in the manuscript according to the reviewer's suggestions.

Specific Points:

1. **The reviewer stated:** *"Line 185. The analysis of the expression profiles including PD-L1 was not shown in the manuscript."*

Response: We thank the reviewer for pointing out this issue. We have included the data in Fig. R1 below (also shown in supplementary Fig.3b of the revised manuscript). Based on the volcano plot of differentially expressed genes in U2OS and U2OS p53-null cells, PD-L1 is listed as a major target of VPRBP regardless of p53 status.

Fig. R1. Volcano plot of differentially expressed genes in U2OS and U2OS p53-null cells. Green dot: down-regulated gene; Blue dot: up-regulated gene; Gray dot: no-change gene.

2. **The reviewer stated:** *"In Figures 4 and 7, PD-L1 IHC staining is required to demonstrate its upregulation in vivo in tumors (potentially in peripheral tissues as well to demonstrate safety despite enhanced expression in the periphery)"*

Response: This is an excellent point. Following the suggestion by the reviewer, we performed the IHC and FACS analysis of PD-L1. As shown in Fig. R2 below (also shown in Supplementary fig. 6e-h and fig. 11d of the revised manuscript), the levels of the PDL1 were upregulated in the tumors upon VPRBP knockdown by FACS analysis (Fig. R2a-c). Moreover, by the IHC staining with the anti-PDL1 antibody, the levels of PDL1 were significantly upregulated in the tumor samples upon either VPRBP knockdown (Fig. R2d) or USP2 inhibition by ML364 treatment (Fig. R2e). In addition, to follow the reviewer's

comment about the safety in peripheral tissues, we performed the anti-PDL1 IHC staining in mouse spleens with the treatment as indicated in Fig. R2f. Indeed, USP2 inhibition by ML364 increased the expression levels of PD-L1 in spleens, in consistent with the result in tumors. Nevertheless, the H&E staining data demonstrated that no obvious morphology changes in mouse spleens (Fig. R2g) were detected; the same results were also confirmed in other major tissues including liver, heart, kidney, thymus, small intestine and bone marrow to support the safety issue of this treatment (Supplementary Fig. S10).

Fig. R2. a, Representative FACS data of PD-L1 in EMT6 control and shVPRBP tumors. b, Mean fluorescence intensity (MFI) of PD-L1+ tumor cells. c, The percentage of PD-L1+ cells in EMT6 control and shVPRBP tumors. d, Representative immunohistochemistry images of PDL1 staining in EMT6 tumors. Scale bar:25µM. e, Representative immunohistochemistry images of PD-L1 staining in EMT6 tumors with indicated treatment. Scale bar: 25µM. f,

Representative immunohistochemistry images of PD-L1 staining in mouse spleen with the treatment as indicated. Scale bar: 100µM. g, Representative images of Hematoxylin and Eosin (H&E) staining in indicated tissues of Balb/c mice treated with vehicle or 30mg/kg ML364. Scale bar: 100µM.

3. **The reviewer stated:** *“In the discussion sections, the authors need to discuss or speculate the specific mechanism of the synergy of PD-L1 blockade and USP2 inhibition and its specificity to the tumor environment.”*

Response: This is an excellent point. Following the suggestion by the reviewer, we have now included the discussion about the specific mechanism of the synergy of PD-L1 blockade and USP2 inhibition and its specificity to the tumor environment as the followings.

(in the discussion section of the revised manuscript)

Our study indicates that VPRBP inhibition produces two seemingly opposing effects on tumor development by suppressing tumor cell growth through activation of p53 but also allowing tumor cells to evade immunosurveillance through increasing PD-L1 levels. Indeed, USP2 inhibition alone partially induces tumor growth suppression but more strikingly, the combination of USP2 inhibition and PD-1/PD-L1 blockade promotes vigorous tumor regression and long-term survival of all tumor-bearing mice. Interestingly, clinical studies demonstrated that the success of PD1–PD-L1 blockade by either anti-PD1 or anti-PD-L1 antibody has a positive correlation with PD-L1 expression levels in tumor cells (PMID: 30527665; PMID: 25428504; PMID: 22658127). Thus, although high levels of PD-L1 prevents cytotoxic T cells from effectively targeting tumor cells, it apparently also serves as a potential selective marker for patient stratification for PD1–PD-L1 blockade therapy. Notably, despite the fact that the immunotherapy has been proved very effective in the treatment of certain types of human cancers, a number of human tumors fail to respond to the immunotherapy. It will be very interesting to examine whether this combination treatment is able to reactivate both p53 function and the immune response particularly, for the tumors previously unresponsive to the immunotherapy alone.

4. **The reviewer stated:** *“In the discussion sections, the authors need to discuss how their new finding link to or distinguish from several other studies that demonstrate regulatory mechanisms of PD-L1 stability by post-translational modifications (i.e. glycosylation, ubiquitination, deubiquitination, and so on).”*

Response: This is an excellent point. Following the suggestion by the reviewer, we have now included the discussion regarding this issue as the followings.

(in the discussion section of the revised manuscript)

Recent studies indicate that the levels and functions of PD-L1 are tightly regulated by several other cellular factors through different mechanisms including glycosylation, ubiquitination, deubiquitination, and acetylation (PMID: 29160310; PMID: 27572267; PMID: 27866850; PMID: 32839551). Nevertheless, the USP2-VPRBP axis is able to target both PD-L1 and p53 simultaneously. Thus, our study not only provides a new layer

of PD-L1 regulation but also has significant implications for the treatment of the tumors that retain wild type p53 to improve the therapeutic efficacy.

5. **The reviewer stated:** *"Figure number seems incorrect in line 511 "Supplementary Fig.8"*

Response: The reviewer was right. We have corrected the typo accordingly.

6. **The reviewer stated:** *"Recommend reading through for grammar mistakes to enhance current readability."*

Response: We sincerely apologize for that. We have now asked several people to read through to correct those mistakes in the text.

#RE: NCOMMS-22-29249

Authors Response to Reviewer #2:

We thank the positive comments on our study by Reviewer #2. We also appreciate the reviewer for raising the issue about some of the “unnecessary” strong statements. As described below, we have taken these points very seriously and addressed them according to the comments by the reviewer. Some of the statements have been either removed or revised as the reviewer suggested. The title of the manuscript is also modified according to the reviewer’s suggestion. The old working model (original Fig. 13) is also removed in the revised version as the reviewer suggested.

- 1. The reviewer stated:** *“A potential issue the authors need to address is the over interpretation of their results. In addition to reporting the findings, the authors made a strong statement that drugging VPRBP/USP2 is an alternative way of activating p53 without affecting the MDM2 feedback loop, thus is superior to MDM2 inhibitors that are known to have toxicities. There are published studies suggesting that knockout of VPRBP/DCAF1 or USP2 activate p53 through affecting MDM2, possibly because DCAF1 regulates ribosomal biogenesis that is linked to MDM2 regulation (PMID: 33355139). Therefore, the strong p53 activation after targeting VPRBP and USP2 are likely caused by disruption of the MDM2 feedback loop. The authors’ own result showing MDM2 overexpression still degrades p53 after USP2 inactivation does not mean that the MDM2 loop is fully functional.”*

Response: This point is well taken. Following the suggestion from the reviewer, we have performed additional experiments about the possible other interpretation.

The study from Yue Xiong lab (PMID: 33355139) reported that VPRBP knockdown in U2OS cells that stably expressed F-MDM2 increased the interaction of MDM2 and ribosomal protein RPL11, thus leading to p53 stabilization. To examine if this is the case in our system, we treated H1299 cells that stably expressed F-MDM2 with ML364 for 48h, and then determined RPL11 in F-MDM2 immunoprecipitates.

Fig. R1. USP2 inhibition by ML364 has no obvious effect on the RPL11 and Mdm2 interaction. F-MDM2 H1299 cells were treated with or without 10 μ M of ML364 for 48h. After anti-Flag immunoprecipitation, PRL11 proteins were detected in eluates by western blot analysis.

As shown in Fig. R1, the treatment of ML364 indeed decreased the VPRBP protein levels as expected while no obvious effect on MDM2 and RPL11 protein levels was detected. Notably, upon the co-immuno-precipitation assays, although we were able to validate the interaction between Mdm2 and RPL11; surprisingly, in contrast to the case in VPRBP-depleted cells as previously reported, we found that no obvious effect on the Mdm2-

RPL11 interaction was detected upon the treatment of USP2 inhibition. It is possible that the effect on VPRBP function by USP2 inhibition is not equal to the effect caused by VPRBP knockdown. For example, USP2 inhibition only reduces the levels of VPRBP by destabilizing VPRBP proteins whereas VPRBP knockdown is able to largely (if not completely) deplete VPRBP proteins in the cells. Thus, USP2 inhibition does not have the same impact on Mdm2 function as VPRBP depletion (or knockdown).

Taken together, USP2 inhibition apparently has no obvious effect on the p53-Mdm2 feedback loop although it remains possible that depletion of VPRBP might affect the p53-Mdm2 feedback loop.

Nevertheless, we have taken the reviewer's point to the heart; indeed, we have removed some of the statements as pointed out by the reviewer and modified our text significantly.

- 2. The reviewer stated:** *"It is also premature to conclude that USP2 inhibitor is superior to MDM2 inhibitors, since they were not directly compared in mice due to lack of potent mouse MDM2 inhibitors. It is true that USP2 inhibitors may have advantages due to effects on other targets besides p53. One could argue that a lower dose of MDM2 inhibitor will be non toxic but can synergize with anti PD1 therapy even better since it does not cause undesirable increase of PDL1 expression. In fact such synergistic activity has been reported (PMID: 33504557)."*

Response: The reviewer was right. Although we are very confident with the implications of the combination treatment by USP2 inhibition and PD-1 mAb in cancers, we should NOT conclude that USP2 inhibitor is superior to Mdm2 inhibitors. Indeed, we have removed all the unnecessary statements from the text.

- 3. The reviewer stated:** *"Therefore, in my opinion the title of manuscript and some conclusions and model (Fig.S13) are somewhat misleading and overtly strong."*

Response: This point is well taken. Following the suggestion from the reviewer, we have modified the title. The old working model (original Fig. 13) is also removed in the revised version.

Other specific comments:

- 4. The reviewer stated:** *"Fig1d legend is missing"*

Response: We apologize for the mistake. We have now added the figure legend of Fig. 1d.

- 5. The reviewer stated:** *"Fig3h. The ubiquitination assay figure labels and legend are confusing. The figure was labeled "Denaturing", PDL1-SFB was pulled down by FLAG IP, which will not work under denaturing condition. Similar label also appears in Fig5f. FigS5g."*

Response: The reviewer was right. We have corrected the labeling in those figures.

- 6. The reviewer stated:** *"The definition of an intact MDM2 feedback loop is not clear and lacks experimental support. Presumably it means that p53 is able to induce MDM2 expression and the*

MDM2 is fully capable of inhibiting p53. Showing MDM2 overexpression being able to degrade p53 in the absence of USP2 is not sufficient for the conclusion (Fig6g)."

Response: This is another excellent point; an intact Mdm2-p53 feedback loop contains two folds: 1) p53 is able to transcriptionally induce Mdm2 expression. 2) Mdm2 can effectively degrade/repress p53. In our studies, we have showed the followings:

Experiments in human cancer cells with USP2 inhibitors:

- (1) The levels of endogenous Mdm2 were activated in a p53-dependent manner upon USP2 inhibition (Main Fig. 6a-c).
- (2) Unlike Mdm2 inhibition, USP2 inhibition has no effect on suppressing Mdm2-mediated p53 degradation (Main Fig. 6d).

Experiments in USP2-null cells;

- (1) The levels of endogenous Mdm2 were induced upon p53 expression in USP2-null cells, suggesting that loss of USP2 expression has no effect on p53-mediated activation of Mdm2(Main Fig. 6g).
- (2) p53 is effectively degraded by Mdm2 expression in USP2-null cells, suggesting that loss of USP2 expression has no effect on Mdm2-mediated degradation of p53(Main Fig. 6f).

These two sets of the data indicate that 1) p53 is able to transcriptionally induce Mdm2 expression upon USP2 inhibition. 2) Mdm2 retains its ability to effectively degrade/repress p53 in the presence of USP2 inhibition. These data indicate that the Mdm2-p53 feedback loop is not affected by USP2 inhibition.

To further support this notion, we performed additional experiments to examine whether endogenous Mdm2 retains its ability to degrade p53 in the presence of USP2 inhibition. To this end, we treated the control and Mdm2 knockdown cells with ML364, respectively.

As expected, the levels of p53 were increased upon the treatment of ML364 but Mdm2 depletion by Mdm2 siRNA was able to further increase p53 levels significantly in those cells, suggesting that Mdm2 is still functional in degrading p53 in the presence of the ML364 inhibitor (Fig.R2).

Fig. R2. Mdm2 retains its ability to degrade p53 in the presence of USP2 inhibition. Western blot analysis for p53 and MDM2 in U2OS cells that were transfected with control (Ctrl.) or MDM2 siRNA followed by 10 μ M of ML364 treatment for 24h.

Taken together, these data demonstrate that by USP2 inhibition can activate p53 function without significantly disrupting the Mdm2-p53 feedback loop.

Nevertheless, we have taken the reviewer's point to the heart; indeed, we have removed these strong statements as pointed out by the reviewer and modified our text significantly.

7. The reviewer stated: “In Fig.4a, VPRBP knockdown induced p53 strongly while also inducing MDM2. This effect is quite similar to DNA damage or ribosomal stress where MDM2 is functionally suppressed despite being induced by p53. This is consistent with previous reports that VPRBP disruption affects ribosomal biogenesis, which then affects MDM2 activity. The authors stated in their introduction that VPRBP regulates p53 through MDM2-independent mechanism (line 153). This is not a fair representation of the literature, given the report from Yue Xiong lab (PMID: 33355139). Previous work also described USP2 inhibition leads to p53 activation through decreasing MDM2 expression (PMID: 27351221).”

Response: This point is almost the same issue raised in Point #1 (see above). Following the suggestion from the reviewer, we have performed additional experiments about the possible other interpretation.

The study from Yue Xiong lab (PMID: 33355139) reported that VPRBP knockdown in U2OS cells that stably expressed F-MDM2 increased the interaction of MDM2 and ribosomal protein RPL11, thus leading to p53 stabilization. To examine if this is the case in our system, we treated H1299 cells that stably expressed F-MDM2 with ML364 for 48h, and then determined RPL11 in F-MDM2 immunoprecipitates.

Fig. R1. USP2 inhibition by ML364 has no obvious effect on the RPL11 and Mdm2 interaction. F-MDM2 H1299 cells were treated with or without 10 μ M of ML364 for 48h. After anti-Flag immunoprecipitation, RPL11 proteins were detected in eluates by western blot analysis.

As shown in Fig. R1, the treatment of ML364 indeed decreased the VPRBP protein levels as expected while no obvious effect on MDM2 and RPL11 protein levels was detected. Notably, upon the co-immuno-precipitation assays, although we were able to validate the interaction between Mdm2 and RPL11; surprisingly, in contrast to the case in VPRBP-depleted cells as previously reported, we found that no obvious effect on the Mdm2-RPL11 interaction was detected upon the treatment of USP2 inhibition. It is possible that the effect on VPRBP function by USP2 inhibition is not equal to the effect caused by VPRBP knockdown. For example, USP2 inhibition only reduces the levels of VPRBP by destabilizing VPRBP proteins whereas VPRBP knockdown is able to largely (if not completely) deplete VPRBP proteins in the cells. Thus, USP2 inhibition does not have the same impact on Mdm2 function as VPRBP depletion (or knockdown).

Moreover, as the reviewer mentioned, a previous study (PMID: 27351221) reported that USP2 knockdown by siRNA reduced MDM2 expression levels upon Nutlin treatment, implicating that USP2 may be also involved in stabilizing Mdm2. As shown in the manuscript (Main Fig. 5j and 6e), by using USP2-null cells, we demonstrated that MDM2

protein levels and half-life were not affected by USP2 depletion. Notably, several different Dubs have been implicated in stabilizing Mdm2 by a similar mechanism. For example, we and others have validated that USP7 acts as a major factor for stabilizing Mdm2 through deubiquitination in both human cancer cells and USP7 knockout mice. Thus, it is very likely that USP2-mediated effect on Mdm2 through deubiquitination is either minor or redundant.

Again, we have taken the reviewer's point to the heart and removed the strong statements as pointed out by the reviewer and modified our text significantly.

#RE: NCOMMS-22-29249

Authors Response to Reviewer #3:

We greatly appreciate the positive view on our study by Reviewer #3. We have taken all the issues raised from the reviewer #3 very seriously and addressed each point with detailed explanation and a large amount of new data. The manuscript is also significantly modified according to the reviewer's suggestions.

Major:

- 1. The reviewer stated:** "Are other TFs known to regulate PD-L1 such as Myc involved in the effect of VPRBP on PD-L1?"

Response: This is an excellent point and we apologized for not explaining this point more clearly in the last version. Indeed, before we focused on the IRF1-VPRBP interaction for PD-L1 regulation, we were also explored the possibility between VPRBP and other well-known factors involved in PD-L1 expression, including c-Myc and STAT3. To this end, we co-transfected H1299 cells with expression vectors encoding IRF1, or STAT3, or c-Myc and VPRBP. As shown in Fig. R1a-c below, VPRBP was readily detected in the immunoprecipitated complexes with IRF1; however, VPRBP was undetectable in the immunoprecipitated complexes with either STAT3 or c-MYC under the same conditions. These data indicate that VPRBP specifically interacts with IRF1, but not with either STAT3 or c-Myc. It also suggests that neither STAT3 nor c-MYC is involved in the effect of VPRBP on PD-L1.

Fig.R1. VPRBP specifically interacts with IRF1, but not with either STAT3 or c-Myc. a, Western blot analysis for VPRBP after immunoprecipitation (IP) of SFB-IRF1, with streptavidin beads, from H1299 cells transfected with VPRBP alone or VPRBP and SFB-IRF1. b, Western blot analysis for VPRBP after immunoprecipitation (IP) of F-STAT3, with anti-Flag beads, from H1299 cells transfected with VPRBP alone or VPRBP and F-STAT3. c, Western blot analysis for VPRBP after immunoprecipitation (IP) of F-c-MYC, with anti-Flag beads, from H1299 cells transfected with VPRBP alone or VPRBP and F-c-MYC.

2. The reviewer stated: “Fig. 2D-2E: It is puzzling why the interaction was demonstrated in A549 and 293 cells while the regulation of VPRBP in PD-L1 are unclear.”

Response: This is another excellent point and we apologized for making the reviewer puzzled in the last version. Since a number of the experiments about VPRBP-mediated regulation of PD-L1 were performed in H1299 cells, to make it more consistent, we have also performed the same co-immuno-precipitation experiments for endogenous VPRBP and IRF1 in H1299 cells. As shown in Fig. R2a below, endogenous IRF1 was readily detected in the immune-complexes precipitated with the VPRBP antibody but not with the control IgG. Conversely, endogenous VPRBP was readily detected in the immune-complexes precipitated with the IRF antibody but not with the control IgG (Fig. R2b). We have replaced the Fig.2d-2e accordingly with the new data and moved the old data performed in A549 and 293 cells to supplementary data section as additional validations.

the Fig.2d-2e accordingly with the new data and moved the old data performed in A549 and 293 cells to supplementary data section as additional validations.

Fig. R2. The co-immunoprecipitation assay for endogenous VPRBP and IRF1 in H1299 cells. a, Western blot

analysis for endogenous IRF1 after immunoprecipitation of endogenous VPRBP in H1299 cells. b, Western blot analysis for endogenous VPRBP after immunoprecipitation of endogenous IRF1 in H1299 cells.

3. The reviewer stated: “Fig. 3: To demonstrate the importance of VPRBP-CUL4-DPPI in PD-L1 stability, it is essential to demonstrate if the ΔE3 mutant of VPRBP affects PD-L1 protein stability.”

Response: This point is well taken. Following the suggestion from the reviewer, we have performed the requested experiment. To this end, we co-transfected the cells with expression vectors encoding PD-L1 alone, either with VPRBP, or with the ΔE3 mutant of

PD-L1. As shown in Fig. R3 below, PD-L1 was effectively degraded by wild-type VPRBP; however, this effect was completely abrogated with the ΔE3 mutant of VPRBP under the same conditions. These data indicate that VPRBP induces PD-L1 degradation through recruiting the CUL4-E3 ligase complex.

Fig. R3. PD-L1 degradation by VPRBP and the ΔE3 mutant. Western blot analysis for PD-L1 in HEK293 cells that were transfected with F-PD-L1 alone, or F-PD-L1 plus FH-VPRBP wildtype (WT) or the ΔE3 mutant for 24h.

4. **The reviewer stated:** *“To gain further insight into the relative contribution of immune system in the antitumor effect mediated by VPRBP KD, it would be informative to compare the antitumor tumor effects on immunodeficient and syngeneic tumor models.”*

Response: Again, we appreciate the comments. This point is well taken. We indeed have compared the antitumor effects of VPRBP KD in immunodeficient (nu/nu) and immunoprecise (Balb/c) mice in the text **page 10** as the followings:

Curiously, VPRBP inhibition potentially produces two opposing effects on tumor development by suppressing tumor cell growth by activation of p53 (Fig. 4a-c) while also allowing tumor cells to evade immunosurveillance through increased PD-L1 levels (Fig. 1-3). Thus, it may be possible to unleash the full therapeutic potential of VPRBP inhibition through immune checkpoint blockade. As expected, in addition to activating the p53 pathway, VPRBP depletion also markedly induced PD-L1 expression levels in both native and p53-null EMT6 cells (Fig. 4a), and similar results were obtained in p53-null 4T1 mouse mammary tumor cells (Supplementary Fig.6a). These VPRBP-mediated effects on PD-L1 levels were further validated by FACS and qPCR analyses using four independent shRNAs against VPRBP (Supplementary Fig.6b-d). Clinical studies have shown that the success of PD1-PD-L1 checkpoint blockade with either anti-PD1 or anti-PD-L1 antibody correlates positively with PD-L1 expression levels on the tumor cells^{52, 61, 62, 63}. Thus, we also examined the impact of VPRBP inhibition on the growth of EMT6 tumor xenografts in immunocompetent (i.e., Balb/c) mice. As shown in Fig. 4e and 4f, VPRBP depletion reduced tumor growth in a small subset of Balb/c mice (2 of 11 mice; Fig. 4e, panel II). This effect is much less pronounced than the uniform reduction of tumor growth observed in nude mice (6 of 6 mice; Fig. 4b), likely reflecting the ability of VPRBP depletion to upregulate PD-L1 expression in immunocompetent mice.

5. **The reviewer stated:** *“CD4 and CD8 depletion assays are needed to strengthen which (or both) is/are the primary effector cells underlying the effect of VPRBP KD and anti-PD-1 combination.”*

Response: This is another excellent point. Following the suggestion from the reviewer, we have performed the requested experiment. To this end, we set up the same combination treatment (Figure R4a) on those tumor models into four groups: Group1 with **IgG control** treatment, Group 2 with normal treatment of **ML364 + PD1mAb**, Group 3 with the treatment of **ML364 + PD1mAb upon CD4 depletion** and Group 4 with the treatment of **ML364 + PD1mAb upon CD8 depletion**. For CD4 and CD8 depletion assays, we first injected 200µg/ml of mouse CD4- or CD8-specific antibodies to those mice (Group 3 and Group 4, respectively), one day before the combination treatment. As expected, FACS analysis of mouse splenocytes demonstrated that CD4 or CD8 T cells were indeed depleted in the mice from group 3 and group 4, respectively (Figure R4b). As expected, the combination treatment of ML364 and PD-1 antibodies effectively repressed tumor growth (Group 2 (ML364 + PD1mAb) vs. Group 1 (IgG) Figure 4c).

Interestingly, upon the deletion of either CD4 or CD8 cells in the same treatment by ML364 and PD1mAb on those mice, loss of CD8 cells completely abrogated the effect of tumor growth suppression whereas depletion of CD4 cells failed to show any effect. These data demonstrate that CD8+ T cells, but not CD4+ T cells, are the primary effector cells underlying the combination treatment in these tumor models.

Fig. R4. Depletion of CD4 or CD8 T cells in EMT6-implanted mice treated with the combination therapy. a, Treatment timeline for b-d. b, Mice were treated with antibodies as shown in a. FACS analysis of CD4+ (Left) and CD8+(Right) T cells in mouse splenocytes. c, IVIS imaging of EMT6 tumors with the treatment as indicated. d, Tumor growth curve of EMT6 tumors with the treatment as indicated. n=10. **** p<0.0001.

6. The reviewer stated: *“The authors showed that VPRBP has a dual effect. One is to stabilize p53 for tumor suppression; the other is to induce PD-L1 expression that presumably suppresses T cell activity. It is unclear what is the primary mechanism underlying the effect of VPRBP in the syngeneic tumors? The authors need to reconcile these two potentially opposing mechanisms. It is puzzling why VPRBP KD does not affect CD4 or CD8 T cell populations given its effect on PD-L1 expression. Does VPRBP KD affect actives of CD4 and/or CD8 T cells?”*

Response: We think that these two opposing effects occur concurrently. On the one hand, VPRBP inhibition leads to p53 stabilization and consequently suppresses tumor growth. This is the good part for tumor suppression. On the other hand, it also unexpectedly upregulates PD-L1 expression to suppress T cell anti-tumor immunity. This is not ideal for tumor suppression, like a bad side effect. We therefore combined the USP2 inhibitor (for downregulating VPRBP levels) with PD1/PDL1 checkpoint blockade to eliminate the bad side effect and maximize the tumor suppression function.

We have now discussed this issue in the revised text as the followings:

Our study indicates that VPRBP inhibition produces two seemingly opposing effects on tumor development by suppressing tumor cell growth through activation of p53 but also allowing tumor cells to evade immunosurveillance through increasing PD-L1 levels. Indeed, USP2 inhibition alone partially induces tumor growth suppression but more strikingly, the combination of USP2 inhibition and PD-1/PD-L1 blockade promotes vigorous tumor regression and long-term survival of all tumor-bearing mice. Interestingly, clinical studies demonstrated that the success of PD1–PD-L1 blockade by either anti-PD1 or anti-PD-L1 antibody has a positive correlation with PD-L1 expression levels in tumor cells (PMID: 30527665; PMID: 25428504; PMID: 22658127). Thus, although high levels of PD-L1 prevents cytotoxic T cells from effectively targeting tumor cells, it apparently also serves as a potential selective marker for patient stratification for PD1–PD-L1 blockade therapy. Notably, despite the fact that the immunotherapy has been proved very effective in the treatment of several types of human cancers, many tumors fail to respond to the immunotherapy. It will be very interesting to examine whether this combination treatment is able to reactivate both p53 function and the immune response particularly, for the tumors previously unresponsive to the immunotherapy alone.

Second, the reviewer also asked *why VPRBP KD does not affect CD4 or CD8 T cell populations given its effect on PD-L1 expression.*

The tumors in main Figure 4g-h were collected at a relative late stage (days 30-40) because we aimed to display the high levels of activated cytotoxic TILs triggered by the combination therapy. Indeed, we were able to demonstrate that the combination of anti-PD-1 treatment and VPRBP depletion induced a marked increase in TILs, including CD4⁺, CD8⁺/granzyme B⁺ T cells (Fig. 4g-h).

If we understood correctly about the reviewer's question, the reviewer meant why we were not able to detect high levels of CD4 or CD8 T cell populations in the cells with VPRBP knockdown alone given its effect on PD-L1 expression. As discussed in our text, high levels of PD-L1 were induced upon VPRBP knockdown. Thus, we should expect high levels of CD4 or CD8 T-cell exhaustion upon PD-L1 upregulation occurred in those tumors. After consulting with several experts in this research area, we were told that in order to detect the levels of CD4 or CD8 T-cell exhaustion in respond to PD-L1 upregulation, we should examine the levels of the CD4+PD1+ or CD8+PD1+ double positive T-cells, at a relative early stage.

Thus, to further address reviewer's question, we need to perform additional experiments to examine the levels of the CD4+PD1+ or CD8+PD1+ double positive T-cells, at a relative early stage. To this end, we collected EMT6 control and shVPRBP tumors at an early stage (days 9-10) after inoculation, and measured the expression of exhaustion marker PD-1 on CD8 T cells by FACS analysis. It is well-characterized that PD-L1 (ligand) on tumor cells recognizes PD-1 (receptor) on cytotoxic CD8 T cell, and then triggers co-inhibitory signal to suppress the cytotoxicity of CD8 T cell, termed T cell exhaustion (PMID: 16382236; PMID: 12218188; PMID: 26086965). Exhausted CD8 T cells express high level of inhibitory receptor PD-1 (exhaustion marker). Despite the levels of CD8 T cells in control and shVPRBP tumors showed no obvious difference (Fig. R5a), significant higher levels of CD8+PD1+ T cells but not CD4+PD1+ T cells in shVPRBP tumors were observed (Fig. R5b-c). These data suggest that high levels of CD8 T cell exhaustion occur in shVPRBP tumors upon PD-L1 upregulation. These data are also consistent with the above observation showing that CD8 T cells, but not CD4 T cells, are the primary effector cells underlying the combination treatment in these tumor models (Fig. R4).

Fig.R5. FACS analysis of CD8⁺PD1⁺ and CD4⁺PD1⁺ TILs in control and shVPRBP EMT6 tumors. a, Percentage of CD8⁺ TILs in control and shVPRBP EMT6 tumors. b, FACS analysis of CD8⁺PD1⁺ TILs in control and shVPRBP EMT6 tumors. c, Mean fluorescence intensity of CD8⁺PD1⁺ TILs in control and shVPRBP EMT6 tumors. ** p<0.01.

7. The reviewer stated: *“It would be important to compare the effects of MDM2i and USP2i on endogenous p53 expression in p53-WT cells, particularly in those EMT6 and RM1 cells, used for tumor growth assays.”*

Response: This is another excellent point. Following the suggestion from the reviewer, we have performed the requested experiment. To this end, we examined endogenous p53 levels after treated both EMT6 and RM1 mouse cells with the same concentration (10 μ M) of the USP2 inhibitor (ML364) and a well-known Mdm2 inhibitor (Nutlin 3a) for the same period of time. The endogenous p53 expression levels were then measured by western blot analysis. As shown in Figure R6 below. Similar levels of p53 proteins were induced by the treatment of either ML364 or Nutlin 3a in both EMT6 and RM1 mouse cells, respectively. These data further validate that p53 was indeed activated upon USP2 inhibition in these mouse tumor cells.

Fig.R6. Western blot analysis for p53 in EMT6 and RM1 cells treated with or without 10 μ M of ML364 and Nutlin 3a for 48h.

8. The reviewer stated: “It would be critical to compare the effects of $\Delta E3$ and ΔAD mutants of VPRBP KD on tumor growth and the response to anti-PD-1 therapy.”

Response: This is another interesting/but very complex question. As mentioned in our text, VPRBP is involved in regulating PD-L1 expression through two different mechanisms. On one hand, VPRBP suppresses IRF1-mediated transcriptional activation of PD-L1; on the other hand, VPRBP directly interact with PD-L1 and induces ubiquitination–mediated degradation of PD-L1. Of note, VPRBP interacts with IRF1 and PD-L1 through different domains. Thus, although the $\Delta E3$ VPRBP mutant is defective in degrading PD-L1, it still retains the ability to interact with IRF1 and repress IRF1-mediated transcriptional activation of PD-L1. However, the ΔAD VPRBP mutant is defective in repressing IRF1-mediated transcriptional activation of PD-L1 but still retains the ability to induce PD-L1 degradation.

As shown in Figure R7a below, by using PD-L1 degradation assays, we found that the $\Delta E3$ VPRBP mutant is indeed defective in degrading PD-L1 but the ΔAD VPRBP mutant fully retains the ability to induce PD-L1 degradation. Conversely, by using IRF1-mediated transactivation assays, we found that the ΔAD VPRBP mutant is defective in repressing IRF1-mediated transcriptional activation of PD-L1 but the $\Delta E3$ VPRBP mutant fully retains the ability to repress IRF1-mediated transcriptional activation of PD-L1 (Figure R7b, 7c). Moreover, according to the previous studies from our lab and others (PMID: 33789902; PMID: 22184063; PMID: 26728942), p53 function is also regulated by VPRBP through both ubiquitination-mediated degradation and transcriptional repression. Thus, both the $\Delta E3$ mutant and the ΔAD VPRBP mutant lose a part of the VPRBP activity in repressing its targets (PD-L1 and

p53); however, both of them also retain a part of the function in regulating tumor growth and the response to anti-PD-1 therapy (Fig. R7d).

Thus, neither of these two VPRBP mutants is truly a loss of function mutant suitable for examine tumor growth and the response to anti-PD-1 therapy.

Fig. R7. The effect of VPRBP $\Delta E3$ and ΔAD mutants. a, Western blot analysis for PD-L1 in HEK293 cells that were transfected with F-PD-L1 alone, or F-PD-L1 plus FH-VPRBP wildtype (WT), $\Delta E3$ or ΔAD mutant for 48h. b and c, Luciferase activity of PDL1-luc in H1299 cells transfected with indicated constructs for 24h. **** $p < 0.0001$; ** $p < 0.01$; ns $p > 0.05$.

Minor point:

- 1. The reviewer stated:** *“The manuscript focuses on the dual effects of USP2-VPRBP pathway. However, the title did not fully represent the body of the work. A title amendment is suggested.*

Response: This point is well taken. Following the suggestion from the reviewer, we have modified the title.

Reviewers' Comments:

Reviewer #1:

Remarks to the Author:

The authors addressed all questions and improved the quality of data in the revised manuscript.

Reviewer #2:

Remarks to the Author:

The revised manuscript by Yi et al has addressed most of my concerns and the paper is improved by additional data and text revisions.

The authors may consider adding their interpretation on why USP2 inhibition is not toxic compared to p53 inhibition in vivo. Implicit from this observation is that p53 activated by USP2 inhibition is functionally different from p53 after MDM2 inhibition, but there is no discussion to explain the difference. In their p53 response data Figure R6, both inhibitors induced similar levels of p53. USP2 inhibition induces same target genes p21/MDM2/PUMA in western blots similar to one would get using MDM2 inhibitors. Presumably the magnitude of p53 activation in vivo, p53 target expression profile, dynamics, or p53-independent functions of USP2 may all play a role in toxicity level. Otherwise one can argue that using low dose MDM2 inhibitor can achieve the same or better result than USP2 inhibitor.

Reviewer #4:

Remarks to the Author:

The authors have been providing response to the referee's comments by including text changes in the manus and putting forward additional figures.

More in details the newly assigned reviewer has been focusing on the comments from the reviewer 3 and the response to each issue.

After careful reviewing, there is no doubt that the authors could reply fully to all of the comments raised by reviewer #3 and the data incorporated in the revised version is indeed convincing.

#RE: NCOMMS-22-29249

Authors Response to Reviewer #1:

1.The reviewer stated: *“The authors addressed all questions and improved the quality of data in the revised manuscript.”*

Response: We thank the reviewer for the comments, which are very helpful to improve the quality of our manuscript.

Authors Response to Reviewer #2:

1.The reviewer stated: *“The revised manuscript by Yi et al has addressed most of my concerns and the paper is improved by additional data and text revisions.”*

The authors may consider adding their interpretation on why USP2 inhibition is not toxic compared to p53 inhibition in vivo. Implicit from this observation is that p53 activated by USP2 inhibition is functionally different from p53 after MDM2 inhibition, but there is no discussion to explain the difference. In their p53 response data Figure R6, both inhibitors induced similar levels of p53. USP2 inhibition induces same target genes p21/MDM2/PUMA in western blots similar to one would get using MDM2 inhibitors. Presumably the magnitude of p53 activation in vivo, p53 target expression profile, dynamics, or p53-independent functions of USP2 may all play a role in toxicity level. Otherwise one can argue that using low dose MDM2 inhibitor can achieve the same or better result than USP2 inhibitor.”

Response: We thank the reviewer very much for his/her thoughtful comments and suggestions during the whole review process. We followed reviewer’s comments to add our interpretation on the difference between USP2 inhibition and MDM2 inhibition in “Discussion” section as following.

“The precise mechanism by which USP2 inhibition is not toxic compared to Mdm2 inhibition *in vivo* remains to be further elucidated. Although p53 can be activated robustly under both conditions, in contrast to Mdm2 inhibition, USP2 inhibition neither affects Mdm2 induction by p53 nor suppresses Mdm2-mediated repression of p53. It is very likely that USP2 inhibition does not significantly affect the negative p53-Mdm2 feedback loop. Previous studies indicate that activation of p53 alone does not automatically result in severe toxicity as observed in the “Super-p53” mice^{28,29} but the negative p53-Mdm2 feedback loop is critical for normal cell homeostasis^{27,31}. Thus, keeping the negative p53-Mdm2 feedback loop intact is likely the major factor to circumvent the toxicity issue upon USP2 inhibition. Moreover, numerous studies indicate that USP2 is able to regulate other cellular factors potentially involved in tumorigenesis⁶⁵. Future studies are clearly needed to examine whether p53-independent functions of USP2 may also contribute to the toxicity level as well as the effect in tumor growth suppression.”

Authors Response to Reviewer #4:

1.The reviewer stated: *“The authors have been providing response to the referee’s comments by including text changes in the manus and putting forward additional figures.”*

More in details the newly assigned reviewer has been focusing on the comments from the reviewer 3 and the response to each issue.

After careful reviewing, there is no doubt that the authors could reply fully to all of the comments raised by reviewer #3 and the data incorporated in the revised version is indeed convincing.”

Response: We really appreciate the reviewer’s positive comments.